# DISSECT: Disentangled Simultaneous Explanations via Concept Traversals

**Asma Ghandeharioun**
Google Research / MIT
aghandeharioun@google.com

**Been Kim**
Google Research
beenkim@google.com

**Chun-Liang Li, Brendan Jou, Brian Eoff**
Google Research
{chunliang,bjou,beoff}@google.com

**Rosalind W. Picard**
MIT
picard@media.mit.edu

## ABSTRACT

Explaining deep learning model inferences is a promising venue for scientific understanding, improving safety, uncovering hidden biases, evaluating fairness, and beyond, as argued by many scholars. One of the principal benefits of counterfactual explanations is allowing users to explore "what-if" scenarios through what does not and cannot exist in the data, a quality that many other forms of explanation such as heatmaps and influence functions are inherently incapable of doing. However, most previous work on generative explainability cannot disentangle important concepts effectively, produces unrealistic examples, or fails to retain relevant information. We propose a novel approach, DISSECT, that jointly trains a generator, a discriminator, and a *concept disentangler* to overcome such challenges using little supervision. DISSECT generates Concept Traversals (CTs), defined as a sequence of generated examples with increasing degrees of concepts that influence a classifier's decision. By training a generative model from a classifier's signal, DISSECT offers a way to discover a classifier's inherent "notion" of distinct concepts automatically rather than rely on user-predefined concepts. We show that DISSECT produces CTs that (1) disentangle several concepts, (2) are influential to a classifier's decision and are coupled to its reasoning due to joint training (3), are realistic, (4) preserve relevant information, and (5) are stable across similar inputs. We validate DISSECT on several challenging synthetic and realistic datasets where previous methods fall short of satisfying desirable criteria for interpretability and show that it performs consistently well. Finally, we present experiments showing applications of DISSECT for detecting potential biases of a classifier and identifying spurious artifacts that impact predictions.

## 1 INTRODUCTION

Explanation of the internal inferences of deep learning models remains a challenging problem that many scholars deem promising for improving safety, evaluating fairness, and beyond [13, 16, 23, 49]. Many efforts in explainability methods have been working towards providing solutions for this challenging problem. One way to categorize them is by the type of explanations, some post hoc techniques focusing on the importance of individual features, such as saliency maps [19, 47, 76, 79], some on importance of individual examples [33, 34, 39, 87], some on importance of high-level concepts [35]. There has been active research into the shortcomings of explainability methods (e.g. [1, 30, 61, 70, 84]) and determining when attention can be used as an explanation [84].

While these methods focus on information *that already exists* in the data, either by weighting features or concepts in training examples or by selecting important training examples, recent progress in generative models [12, 29, 36, 40, 45] has lead to another family of explainability methods that provide explanations by *generating new* examples or features [14, 32, 68, 73]. New examples or features can be used to generate *counterfactuals* [82] allowing users to ask: what if this sample were to be classified as the opposite class, and how would it differ? This way of explaining mirrors the

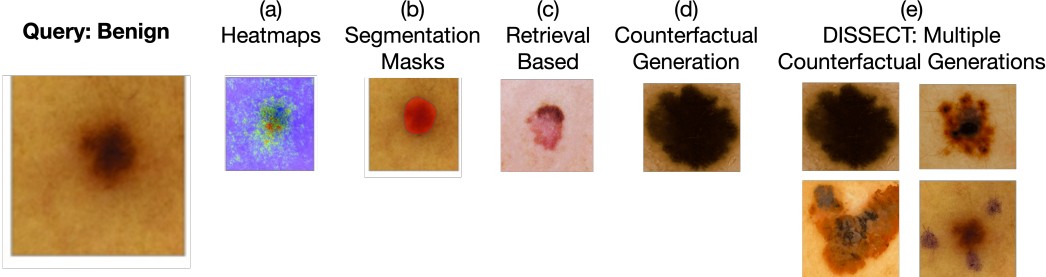

Figure 1: Examples of explainability methods applied to a melanoma classifier. Explanation by (a) heatmaps (e.g. [19, 47, 76, 79]), (b) segmentation masks (e.g. [22, 69]), (c) sample retrieval (e.g. [72]), (d) counterfactual generation (e.g. [68, 73]), (e) and multiple counterfactual generations such as DISSECT. Multiple counterfactuals could highlight several different ways that changes in a skin lesion could reveal its malignancy and overcome some of the blind spots of the other approaches. For example, they can demonstrate that large lesions, jagged borders, and asymmetrical shapes lead to melanoma classification. They can also show potential biases of the classifier by revealing that surgical markings can spuriously lead to melanoma classification.

way humans reason, justify decisions [3], and learn [4, 6, 83]. Additionally, users across visual, auditory, and sensor data domains found examples the most preferred means of explanations [31].

To illustrate the benefits of counterfactual explanations, consider a dermatology task where an explanation method is used to highlight why a certain sample is classified as benign/malignant (Fig. 1). Explanations like heatmaps, saliency maps, or segmentation masks only provide partial information. Such methods might hint at what is influential within the sample, potentially focusing on the lesion area. However, they cannot show what kind of changes in color, texture, or inflammation could transform the input at hand from benign to malignant. Retrieval-based approaches that provide examples that show a concept are not enough for answering "what-if" questions either. A retrieval-based technique might show input samples of malignant skin lesions that have similarities to a benign lesion in patient A, but from a different patient B, potentially from another body part or even a different skin tone. Such examples do not show what this benign lesion in patient A would have to look like if it were classified as malignant instead. On the other hand, counterfactuals depict *how* to modify the input sample to change its class membership. A counterfactual explanation visualizes what a malignant tumor could look like in terms of potential color or texture, or inflammation changes on the skin. Better yet, multiple counterfactuals could highlight several different ways that changes in a skin lesion could reveal its malignancy. For a classifier that relies on surgical markings [86] as well as meaningful color/texture/size changes to make its decision, a single explanation might fail to reveal this flaw resulting from dependence on spurious features, but multiple distinct explanations shed light on this phenomenon. Multiple explanations are strongly preferred in several other domains, such as education, knowledge discovery, and algorithmic recourse generation [17, 54].

In this work, we propose a generation-based explainability method called DISSECT that generates *Concept Traversals* (CTs)–sequences of generated examples with increasing degrees of concepts' influence on a classifier's decision. While current counterfactual generation techniques fail to satisfy the most consistently agreed-upon properties desired for an explainability method simultaneously [50, 60, 68, 73, 74], DISSECT aims to overcome this challenge. CTs are generated by jointly training a generator, a discriminator, and a CT disentangler to produce examples that (1) express one distinct factor [50] at a time; (2) are influential [73, 74] to a classifier's decision and are coupled to the classifier's reasoning, due to joint training; (3) are realistic [73]; (4) preserve relevant information [50, 60, 68]; and (5) are stable across similar inputs [21, 50, 60]. Compared to other approaches that require a human to identify concepts a priori and find samples representing them, e.g. [9, 35], DISSECT uncovers them automatically. We compare DISSECT with several baselines, some of which have been optimized for disentanglement, some used extensively for explanation, and some that fall in between. DISSECT is the only technique that performs well across all these dimensions. Other baselines either have a hard time with influence, lack fidelity, generate poor quality and unrealistic samples, or do not disentangle properly. We evaluate DISSECT using `3D Shapes` [7], `CelebA` [44], and a new synthetic dataset inspired by real-world dermatology challenges. We show that DISSECT outperforms prior work in addressing all of these challenges. We discuss applications to detect a classifier's potential biases and identify spurious artifacts using simulated experiments.

This paper makes five main contributions: 1) presents a novel counterfactual explanation approach that manifests several desirable properties outperforming baselines; 2) demonstrates applications

through experiments showcasing the effectiveness of this approach for detecting potential biases of a classifier; 3) presents a set of explainability baselines inspired by approaches used for generative disentanglement; 4) translates desired properties commonly referred to in the literature across forms of explanation into measurable quantities for benchmarking and evaluating counterfactual generation approaches; 5) releases a new synthetic dermatology dataset inspired by real-world challenges.

## 2 RELATED WORK

Our method relates to the active research area of post hoc explainability methods. One way to categorize them is by explanation form. While met with criticisms [1, 75], many feature-based explainability methods are shown to be useful [47, 79], which assign a weight to each input feature to indicate their importance in classification [10, 57]. Example-based methods are another popular category [33, 34, 39, 87] that instead assign importance weights to individual examples. More recently, concept-based methods have emerged that attribute weights to concepts, i.e., higher-level representations of features [22, 25, 35] such as "long hair".

Our work leverages recent progress in generative modeling, where the explanation is presented through several conditional generations [52, 56]. Efforts for the "discovery" of concepts are also related to learning disentangled representations [12, 29, 35, 36], which has been shown to be challenging without additional supervision [45]. Recent findings suggest that weak supervision in the following forms could make the problem identifiable: how many factors of variation have changed [71], using labels for validation [46], or using sparse labels during training [46]. While some techniques like Conditional Subspace VAE (CSVAE) [38] began to look into conditional disentanglement by incorporating labels, their performance has not yet reached their unconditional counterparts. Our work uses a new form of weak supervision to improve upon existing methods: the posterior probability and gradient of the classifier-under-test.

Many explainability methods have emerged that use generative models to modify existing examples or generate new examples [14, 15, 26, 32, 48, 68, 73]. Most of these efforts use pre-trained generators and generate a new example by moving along the generator's embedding. Some aim to generate samples that would flip the classifier's decision [15, 32], while others aim to modify particular attribution of the image and observe the classifier's decision change [14]. More recent work allows the classifier's predicted probabilities or gradients to flow through the generator during training [68], and visualize the continuous progression of a key differential feature [73]. This progressive visualization has additional benefits, as Cai et al. [9] showed that pathologists could iterate more quickly using a tool that allowed them to progressively change the value of a concept, compared to purely using examples. Additionally, continuous exaggeration of a concept facilitated thinking more systematically about the specific features important for diagnosis.

Most of these approaches assume that there is only one path that crosses the decision boundary, and they generate examples along that path. Our work leverages both counterfactual generation techniques and ideas from the disentanglement literature to generate diverse sets of explanations via multiple distinct paths influential to the classifier's decision, called Concept Traversals (CT). Each CT is made of a sequence of generated examples that express increasing degrees of a concept.

## 3 METHODS

Our key innovation is developing an approach to provide multiple potential counterfactuals. We show how to successfully incorporate the classifier's signal while *disentangling* relevant concepts. We review existing generative models for building blocks of our approach, and follow Singla et al. [73] for DISSECT's backbone. This choice is informed by the good conditional generation capability of [73] and being a strong interpretability baseline. We then introduce a novel component for unsupervised disentanglement of explanations. Multiple counterfactual explanations could reveal several different paths that flip the classification outcome and shed light on different biases of the classifier. However, a single explanation might fail to reveal such information. Disentangling distinct concepts without relying on predefined user labels reduces the burden on the user side. Additionally, it can surface concepts that the user might have missed otherwise. However, unsupervised disentanglement is particularly challenging [45]. To address this challenge, we design a disentangler network. The disentangler, along with the generator, indirectly enforces different concepts to

be as distinct as possible. This section summarizes our final design choices. For more details about baselines, DISSECT, and ablations studies, see §A.1, §A.2, and §A.6, respectively.

**Notation.** Without loss of generality, we consider a binary classifier $f \colon X \to Y = \{-1, 1\}$ such that $f(x) = p(y = 1|x)$ where $x \sim \mathcal{P}_X$. We want to find $K$ latent concepts that contribute to the decision-making of $f$. Given $x$, $\alpha \in [0, 1]$, and $k$, we want to generate an image, $\bar{x}$, by perturbing the latent concept $k$, such that the posterior probability $f(\bar{x}) = \alpha$. In addition, when conditioning on $x$ and a concept $k$, by changing $\alpha$, we hope for a smooth change in the output image $\bar{x}$. This smoothness resembles slightly changing the degree of a concept. Putting it together, we desire a generic generation function that generates $\bar{x}$, defined as $\mathcal{G}(x, \alpha, k; f) : X \times [0, 1] \times \{0, 1, \ldots, K - 1\} \to X$, where $f(\mathcal{G}(x, \alpha, k; f)) \approx \alpha$. The generation function $\mathcal{G}$ conditions on $x$ and manipulates the concept $k$ such that a monotonic change in the $f(\mathcal{G}(x, 0, k)), \ldots, f(\mathcal{G}(x, 1, k))$ is achieved.

## 3.1 ENCODER-DECODER ARCHITECTURE

We realize the generic (conditional) generation process using an encoder-decoder framework. The input $x$, is encoded into an embedding $E(x)$, before feeding into the generator $G$, which serves as a decoder. In addition to $E(x)$, we have a conditional input $c(\alpha, k)$ for the $k^{\text{th}}$ latent concept we are going to manipulate with level $\alpha$. $\alpha$ is the desired prediction of $f$ we want to achieve in the generated sample by perturbing the $k^{\text{th}}$ latent concept. The generative process $\mathcal{G}$ can be implemented by $G(E(x), c(\alpha, k))$, where we can manipulate $c$ for interpretation via conditional generation.

There are many advances in generative models, which can be adopted to train $G(E(x), c(\alpha, k))$. For example, Variational Autoencoders (VAE) explicitly designed for disentanglement [11, 29, 38, 41]. However, VAE-based approaches often generate blurry images which requires customized architecture designs [63, 81]. In contrast, GANs [24] tend to generate high-quality images. Therefore, we consider an encoder-decoder design with GANs [5, 55, 58, 64, 66, 89] to have flexible controllability and high quality generation for explainability.

There are different design choices for realizing the conditioning code $c$. A straightforward option is multiplying a $K$-dimensional one-hot vector, which specifies the desired concept $k$, with the probability $\alpha$. However, conditioning on continuous variables is usually challenging for conditional GANs compared with their discretized counterparts [52]. Therefore, we discretize $\alpha$ into $[0, 1/N, \ldots, 1]$, and parametrize $c$ as a binary matrix $[0, 1]^{(N+1) \times K}$ for $N + 1$ discrete values and $K$ concepts, following Singla et al. [73]. Therefore, $c(\alpha, k)$ denotes that we set the elements $(m, k)$ to be 1, where $m$ is an integer and $m/N \le \alpha < (m + 1)/N$. See § A.2 for more details.

We include the following elements in our objective function following Singla et al. [73], and further modify them to support *multiple* concepts:

**Interpreting classifiers.** We align the generator outputs with the classifier being interpreted [1]:
$$\mathcal{L}_f(G) = \alpha \log \Big( f\big( G\left(x, c(\alpha, k)\right) \big) \Big) + \Big( 1 - \alpha \Big) \log \Big( 1 - f\big( G\left(x, c(\alpha, k)\right) \big) \Big). \tag{1}$$
**Reconstruction.** We adopt $\ell_1$ reconstruction loss in our encoder-decoder design [55, 89]. Given $x \sim \mathcal{P}_X$ and classifier $f$, the ground-truth $\alpha$ for all concepts is $f(x)$:
$$\mathcal{L}_{\text{rec}}(G) = \frac{1}{K} \sum_{k=0}^{K-1} \Big\| x - G\big(x, c\left(f(x), k\right)\big) \Big\|_1. \tag{2}$$
**Cycle consistency.** Define $\bar{x}_{\alpha,k} = G(x, c(\alpha, k))$. We add a cycle consistency loss [89]:
$$\mathcal{L}_{\text{cyc}}(G) = \frac{1}{K} \sum_{k=0}^{K-1} \mathbb{E}_\alpha \left[ \Big\| x - G\big(\bar{x}_{\alpha,k}, c\left(f(x), k\right)\big) \Big\|_1 \right]. \tag{3}$$
**GAN loss.** We use spectral normalization [53] with its hinge loss variant [52] to compare the distribution $x \sim \mathcal{P}_X$ and the generated data:[2]
$$\mathcal{L}_{\text{cGAN}}(D) = -\mathbb{E}_{x \sim \mathcal{P}_X} \left[ \min \Big( 0, -1 + D(x) \Big) \right] - \mathbb{E}_{x \sim \mathcal{P}_X} \mathbb{E}_{\alpha,k} \left[ \min \Big( 0, -1 - D\big( G\left(x, c(\alpha, k)\right) \big) \Big) \right], \tag{4}$$
$$\mathcal{L}_{\text{cGAN}}(G) = -\mathbb{E}_{x \sim \mathcal{P}_X} \mathbb{E}_{\alpha,k} \left[ D\big( G\left(x, c(\alpha, k)\right) \big) \right]. \tag{5}$$

---

[1]Note that $G$ is trained having access to $f$'s signal. The LogLoss between the desired probability $\alpha$ and the predicted probability of the perturbed image is minimized. Thus, the model supports fidelity by design.

[2]One can use any GAN loss [2, 24, 27, 42, 53, 88]. Generated data comes from uniformly sampling $k$ & $\alpha$.

## 3.2 Enforcing disentanglement

Ideally, we want the $K$ concepts to capture distinctively different qualities influential to the classifier being explained. Though $G$ is capable of expressing $K$ different concepts, the above formulation does not enforce distinctness among them. To address this challenge, we introduce a *concept disentangler* $R$ inspired by the disentanglement literature [8, 11, 12, 29, 38, 41, 43]. Most disentanglement measures quantify proxies of *distinctness* of changes *across* latent dimensions and *similarity* of changes *within* a latent dimension [45]. This is usually achieved through latent code traversal, keeping a particular dimension fixed, or keeping all but one dimension fixed. Thus, we design $R$ to encourage *distinctness* across concepts and *similarity* within a concept.

The $K$ concepts can be viewed as "knobs" that can modify a sample along a conceptual dimension. Without loss of generality, we assume modifying only one "knob" at a time. Given a pair of images $(x, x')$, $R$ tries to predict which concept $k$, $k \in \{0, \dots, K-1\}$, has perturbed query $x$ to produce $x'$. To formalize this, let $\bar{x}_{k,\alpha} = G(x, c(\alpha, k))$ and $\tilde{x}_k = G(\bar{x}_{k,\alpha}, c(f(x), k))$. The loss function is

$$\mathcal{L}_r(G, R) = \mathbb{CE}(e_k, R(x, \bar{x}_k)) + \mathbb{CE}(e_k, R(\bar{x}_k, \tilde{x}_k)), \tag{6}$$

where $\mathbb{CE}$ is the cross entropy and $e_k$ is a one-hot vector of size $K$ for the $k^{\text{th}}$ concept. Note that while the $k^{\text{th}}$ "knob" can modify a sample query with $\alpha$ degree, $\alpha \in [0, 1/N, \dots, 1]$, $R$ tries to solve a multi-class binary prediction problem. Thus, for any $\alpha$, $\alpha \neq f(x)$, $R(x, \bar{x}_k)$ is trained to predict a one-hot vector for the $k^{\text{th}}$ concept. Furthermore, note that no external labels are required to distinguish between concepts. Instead, by jointly optimizing $R$ and $G$, we penalize the generator $G$ if any concept knob results in similar-looking samples to other concept knobs. Encouraging noticeably distinct changes across concept knobs indirectly promotes disentanglement. In summary, the overall objective function of our method is:

$$\min_{G,R} \max_D \lambda_{\text{cGAN}} \mathcal{L}_{\text{cGAN}}(D, G) + \lambda_f \mathcal{L}_f(D, G) + \lambda_{\text{rec}} \mathcal{L}_{\text{rec}}(G) + \lambda_{\text{rec}} \mathcal{L}_{\text{cyc}}(G) + \lambda_r \mathcal{L}_r(G, R). \tag{7}$$

We call our approach DISSECT as it disentangles simultaneous explanations via concept traversals.

## 4 Experiments

**3D Shapes.** We first use `3D Shapes` [7] purely for validation and demonstration purposes due to the controllability of all its factors. It is a synthetic dataset available under Apache License 2.0 composed of 480K 3D shapes procedurally generated from 6 ground-truth factors of variation. These factors are floor hue, wall hue, object hue, scale, shape, and orientation.

**SynthDerm.** Second, we create a new dataset, `SynthDerm` (Fig. 2). Real-world characteristics of melanoma skin lesions in dermatology settings inspire the generation of this dataset [80]. These characteristics include whether the lesion is asymmetrical, its border is irregular or jagged, is unevenly colored, has a diameter more than 0.25 inches, or is evolving in size, shape, or color over time. These qualities are usually referred to as the ABCDE of melanoma [65]. We generate `SynthDerm` algorithmically by varying several factors: skin tone, lesion shape, lesion size, lesion location (vertical and horizontal), and whether there are surgical markings present. We randomly assign one of the following to the lesion shape: round, asymmetrical, with jagged borders, or multicolored (two different shades of colors overlaid with salt-and-pepper noise). For skin tone values, we simulate Fitzpatrick ratings [20]. Fitzpatrick scale is a commonly used approach to classify the skin by its reaction to sunlight exposure modulated by the density of melanin pigments in the skin. This rating has six values, where 1 represents skin with lowest melanin and 6 represents skin with highest melanin. For our synthetic generation, we consider six base skin tones that similarly resemble different amounts of melanin. We also add a small amount of random noise to the base color to add further variety. `SynthDerm` includes more than 2,600 64x64 images.

**CelebA.** We also include the `CelebA` dataset [44] that is available for non-commercial research purposes. This dataset includes "identities [of celebrities] collected from the Internet" [78], is realistic, and closely resembles real-world settings where the attributes are nuanced and not mutually independent. `CelebA` includes more than 200K images with 40 annotated face attributes, such as smiling and bangs.

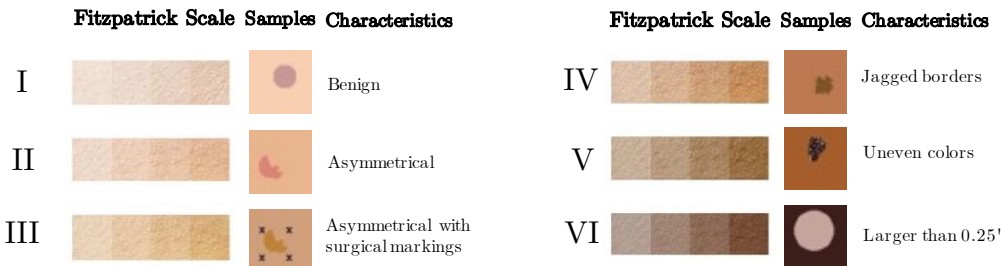

Figure 2: Illustration of `SynthDerm` dataset that we algorithmically generated. Fitzpatrick scale of skin classification based on melanin density and corresponding examples in the dataset are visualized.

## 4.1 BASELINES

We consider several baselines. First, we modify a set of VAEs explicitly designed for disentanglement [8, 11, 29, 41] to incorporate the classifier's signal during their training processes and encourage the generative model to learn latent dimensions that influence the classifier, i.e., learning *Important* CTs. While these approaches are not naturally designed to explain an external classifier, they have been designed to find distinct factors of variation. We refer to these baselines with a **-mod** postfix, e.g. $\beta$-**VAE-mod**. Second, we include **CSVAE** [38] that aims to solve unsupervised learning of features associated with a specific label using a low-dimensional latent subspace that can be independently manipulated. CSVAE, without any modification, is the closest alternative in the literature that tries to discover both *Important* and *Distinct* CTs. Third, we include Explanation by Progressive Exaggeration (**EPE**) [73], a GAN-based approach that learns to generate one series of counterfactual and realistic samples that change the prediction of $f$, given data and the classifier's signal. EPE is the most similar baseline to DISSECT in terms of design choices; however, it is not designed to handle distinct CTs. We introduce **EPE-mod** by modifying EPE to allow learning multiple pathways by making the generator conditional on the CT dimension (more details in §A.1).

## 4.2 EVALUATION METRICS

This section provides a brief summary of our evaluation metrics. We consider *Importance*, *Realism*, *Distinctness*, *Substitutability*, and *Stability*, which commonly appear as desirable qualities in explainability literature.[3] See more in §A.3.

**Importance.** Explanations should produce the desired outcome from the black-box classifier $f$. Previous work has referred to this quality using different terms, such as importance [22], compatibility with classifier [73], and classification model consistency [74]. While most previous methods have relied on visual inspection, we introduce a quantitative metric to measure the gradual increase of the target class's posterior probability through a CT. Notably, we compute the correlation between $\alpha$ and $f(I(x, \alpha, k; f))$ introduced in § 3. For brevity, we refer to $f(I(x, \alpha, k; f))$ as $f(\bar{x})$ in the remainder of the paper. We also report the mean-squared error and the Kullback–Leibler (KL) divergence between $\alpha$ and $f(\bar{x})$. We also calculate the performance of $f$ on the counterfactual explanations. Specifically, we replace the test set of real images with their generated counterfactual explanations and quantify the performance of the pre-trained black-box classifier $f$ on the counterfactual test set. Better performance suggests that counterfactual samples are compatible with $f$ and lie on the correct side of the classifier's boundary. Note that Importance scores can be calculated per concept and provide a ranking across concepts, or they can be aggregated to summarize overal performance. See §A.7 for more information about concept ranking using individial concept Importance scores.

**Realism.** We need the generated samples that form a CT to look realistic to enable users to identify concepts they represent. It means the counterfactual explanations should lie on the data manifold. This quality has been referred to as realism or data consistency [73]. We use Fréchet Inception Distance (FID) [28] as a measure of *Realism*. Additionally, inspired by [18], we train a post hoc classifier that predicts whether a sample is real or generated. Note that it is essential to do this step post hoc and independent from the training procedure because relying on the discriminator's accuracy in an adversarial training framework can be misleading [18].

**Distinctness.** Another desirable quality for explanations is to represent inputs with non-overlapping concepts, often referred to as diversity [50]. Others have suggested similar properties such as co-

---

[3]Eq. 1 encourages Importance, Eq. 5 encourages Realism, Eq. 6 encourages Distinctness, Eq. 2 and Eq. 3 indirectly help with Stability. All the elements of Eq. 7 contribute to improving Substitutability.

Figure 3: Examples from `3D Shapes`. EPE and EPE-mod converge to finding the same concept, despite EPE-mod's ability to express multiple pathways to switch classifier outcomes from False to True. DISSECT discovers the two ground-truth concepts: $CT_1$ flips the floor color to cyan and $CT_2$ flips the shape color to red.

Table 1: Quantitative results on `3D Shapes`. DISSECT performs significantly better or on par with the strongest baselines in all evaluation categories. Modified VAE variants perform poorly in terms of *Importance*, worse than CSVAE, and significantly worse than EPE, EPE-mod, and DISSECT. CSVAE and other VAE variants do not produce high-quality images, thus have poor *Realism* scores; meanwhile, EPE, EPE-mod, and DISSECT generate realistic samples indistinguishable from real images. While the aggregated metrics for *Importance* are useful for discarding VAE baselines with poor performance, they do not show a consistent order across EPE, EPE-mod, and DISSECT. Our approach greatly improves *Distinctness*, especially compared to EPE-mod. EPE is inherently incapable of doing this, and the extension EPE-mod does, but poorly. For the *Substitutability* scores, note the classifier's precision, recall, and accuracy when training on actual data is 100%. (*Some methods only generate samples with $f(\bar{x}) = 0.0$. Correlation with a constant value is undefined.)

| | Importance | | | | | | | Realism | | | Distinctness | | | | | Substitutability | | | Stability | |
|---|---|---|---|---|---|---|---|---|---|---|---|---|---|---|---|---|---|---|---|---|
| | ↑R | ↑ρ | ↓KL | ↓MSE | ↑CF Acc | ↑CF Prec | ↑CF Rec | ↓Acc | ↓Prec | ↓Rec | ↑Acc | ↑Prec (micro) | ↑Prec (macro) | ↑Rec (micro) | ↑Rec (macro) | ↑Acc Sub | ↑Prec Sub | ↑Rec Sub | ↓CF MSE | ↓Prob JSD |
| VAE-mod | 0.1 | 0.3 | inf | 0.42 | 50.0 | 0 | 0 | 94.9 | 92.1 | 99.6 | 86.3 | 95.1 | 95.3 | 80.6 | 80.6 | 14.7 | 14.2 | 69.4 | 0.151 | **0.0000** |
| β-VAE-mod | 0.0 | 0.0 | inf | 0.42 | 50.0 | 0 | 0 | 99.5 | 99.0 | 100 | 90.7 | 92.2 | 92.2 | 91.0 | 91.0 | 42.5 | 8.2 | 19.8 | 0.197 | **0.0000** |
| Annealed-VAE-mod | N/A* | N/A* | inf | 0.42 | 50.0 | 0 | 0 | 100 | 100 | 100 | 34.0 | 53.2 | 49.2 | 1.20 | 1.2 | 35.8 | 14.4 | 48.1 | 0.175 | **0.0000** |
| DIPVAE-mod | 0.1 | 0.5 | inf | 0.41 | 52.3 | 100 | 4.6 | 100 | 100 | 100 | 97.2 | 96.7 | 96.9 | 96.5 | 96.5 | 19.0 | 19.0 | 100 | 0.127 | 0.0001 |
| βTCVAE-mod | 0.5 | 0.7 | inf | 0.42 | 50.0 | 0 | 0 | 100 | 100 | 100 | **100** | **100** | **100** | **100** | **100** | 36.4 | 21.3 | 87.3 | 0.143 | **0.0000** |
| CSVAE | 0.3 | 0.3 | 5.5 | 0.28 | 64.6 | 100 | 29.3 | 100 | 100 | 100 | 71.4 | 74.7 | 76.4 | 87.2 | 87.2 | 47.0 | 23.8 | 81.3 | 19.544 | 0.0274 |
| EPE | 0.8 | **0.8** | 1.54 | 0.09 | 98.4 | 100 | 96.7 | 50.1 | **0** | **0** | - | - | - | - | - | 99.2 | 95.9 | **100** | 0.134 | 0.0004 |
| EPE-mod | **0.9** | 0.7 | 2.2 | **0.08** | **99.7** | 100 | **99.4** | 49.3 | **0** | **0** | 45.3 | 49.6 | 49.8 | 30.3 | 30.3 | 91.0 | **100** | 52.5 | 0.128 | 0.0002 |
| **DISSECT** | 0.8 | **0.8** | 1.61 | **0.08** | 98.7 | 100 | 97.5 | **49.3** | **0** | **0** | **100** | **100** | **100** | **100** | **100** | **100** | 99.7 | **100** | **0.102** | 0.0003 |

herency, meaning examples of a concept should be similar but different from examples of other concepts [22]. To quantify this in a counterfactual setup, we introduce a distinctness measure capturing the performance of a secondary classifier that distinguishes between CTs. We train a classifier post hoc that given a query image $x$, a generated image $x'$, and $K$ number of CTs, predicts $CT_k$ that has perturbed $x$ to produce $x'$, $k \in \{1, \ldots, K\}$.

**Substitutability.** The representation of a sample in terms of concepts should preserve relevant information [50, 60]. Previous work has formalized this quality for counterfactual generation contexts through a proxy called substitutability [68]. Substitutability measures an external classifier's performance on real data when it is trained using only synthetic images.[4] A higher substitutability score suggests that explanations have retained relevant information and are of high quality.

**Stability.** Explanations should be coherent for similar inputs, a quality known as stability [21, 50, 60]. To quantify stability in counterfactual explanations, we augment the test set with additive random noise to each sample $x$ and produce $S$ randomly augmented copies $x_i''$. Then, we generate counterfactual explanations $\bar{x}$ and $\bar{x}_i''$, respectively. We calculate the mean-squared difference between counterfactual images $\bar{x}$ and $\bar{x}_i''$ and the resulting Jensen Shannon distance (JSD) between the predicted probabilities $f(\bar{x})$ and $f(\bar{x}_i'')$.

### 4.3 CASE STUDY I: VALIDATING THE QUALITIES OF CONCEPT TRAVERSALS

Considering `3D Shapes` [7], we define an image as "colored correctly" if the shape hue is red or the floor hue is cyan. We train a classifier to detect whether a sample image is "colored correctly" or not. In this synthetic experiment, only these two independent factors contribute to the decision of this classifier. Given a not "colored correctly" query, we would like the algorithm to find a CT related to the shape color and another CT associated with the floor color–two different pathways that lead to switching the classifier outcome.[5]

---

[4]This metric has been used in other contexts outside of explainability and has been called Classification Accuracy Score (CAS) [62]. CAS is more broadly applicable than Fréchet Inception Distance [28] & Inception Score [67] that are only useful for GAN evaluation.

[5]In this scenario, these two ground-truth concepts do not directly apply to switching the classifier outcome from True to False. For example, if an image has a red shape and a cyan floor, both of these colors need to be changed to switch the classification outcome. We still observe that DISSECT finds CTs that change different combinations of colors, but the baseline methods converge to the same CT. See §A.5.1 for more details.

Figure 4: Examples from `SynthDerm` comparing DISSECT with the strongest baseline, EPE-mod. We illustrate three queries with different Fitzpatrick ratings [20] and visualize the two most prominent concepts for each technique. We observe that EPE-mod converges on a single concept that only vaguely represents meaningful ground-truth concepts. However, DISSECT successfully finds concepts describing asymmetrical shapes, jagged borders, and uneven colors that align with the ABCDE of melanoma [65]. DISSECT also identifies concepts for surgical markings that spuriously impact the classifier's decisions.

Table 2: Quantitative results on `SynthDerm`. DISSECT performs consistently best in all categories. For anchoring the *Substitutability* scores, note that the precision, recall, and accuracy of the classifier when training on actual data is 97.7%, 100.0%, and 95.4%, respectively.

| | Importance | | | | | | | Realism | | | | Distinctness | | | | | Substitutability | | | Stability | |
|---|---|---|---|---|---|---|---|---|---|---|---|---|---|---|---|---|---|---|---|---|---|
| | ↑R | ↑ρ | ↓KL | ↓MSE | ↑CF Acc | ↑CF Prec | ↑CF Rec | ↓FID | ↓Acc | ↓Prec | ↓Rec | ↑Acc | ↑Prec (micro) | ↑Prec (macro) | ↑Rec (micro) | ↑Rec (macro) | ↑Acc Sub | ↑Prec Sub | ↑Rec Sub | ↓CF MSE | ↓Prob JSD |
| CSVAE | 0.25 | 0.64 | 1.78 | 0.12 | 86.7 | 43.9 | 5.2 | 52.6 | 54.6 | 69.9 | 16.3 | 85.1 | 0 | 0 | 0 | 0 | 29.9 | 36.8 | 55.4 | 2.318 | 0.006 |
| EPE | 0.87 | 0.23 | inf | 0.03 | 80.7 | 55.1 | 86.9 | **23.8** | 50.1 | **0** | **0** | - | - | - | - | - | 74.4 | 83.8 | 60.7 | **0.111** | **0.001** |
| EPE-mod | 0.81 | 0.73 | 0.92 | 0.04 | 95.3 | 83.9 | 79.5 | 24.3 | **50.0** | **0** | **0** | 85.1 | 71.1 | 26.3 | 0.7 | 0.7 | 74.3 | 83.5 | 60.7 | 0.239 | 0.002 |
| **DISSECT** | **0.92** | **0.75** | **0.35** | **0.02** | **97.8** | **92.3** | **91.1** | 27.5 | **50.0** | **0** | **0** | **96.0** | **96.5** | **96.5** | **74.6** | **74.6** | **81.0** | **97.2** | **64.0** | 0.338 | **0.001** |

Tab. 1 summarizes the quantitative results on `3D Shapes`. Most VAE variants perform poorly in terms of *Importance*, CSVAE performs slightly better, and EPE, EPE-mod, and DISSECT perform best. Our results suggest that DISSECT performs similarly to EPE that has been geared explicitly toward exhibiting *Importance* and its extension, EPE-mod. Additionally, DISSECT still keeps *Realism* intact. Also, it notably improves the *Distinctness* of CTs compared to relevant baselines.

Fig. 3 shows the examples illustrating the qualitative results for EPE, EPE-mod, and DISSECT. Our results reveal that EPE converges to finding only one of these concepts. Similarly, both CTs generated by EPE-mod converge to finding the same concept, despite being given the capability to explore two pathways to switch the classifier outcome. However, DISSECT finds the two distinct ground-truth concepts through its two generated CTs. This is further evidence for the necessity of the disentanglement loss to enforce the distinctness of discovered concepts. For brevity, three sample queries are visualized (See more in §A.5.1). Note that given the synthetic nature of this dataset, the progressive paths may not be as informative. Thus, we only visualize the far end of the spectrum, i.e., the example that flips the classification outcome all the way. See §4.5 for the progressive exaggeration of concepts in a more nuanced experiment.

## 4.4 CASE STUDY II: IDENTIFYING SPURIOUS ARTIFACTS

A "high performance" model could learn to make its decisions based on irrelevant features that only happen to correlate with the desired outcome, known as label leakage [85]. One of the applications of DISSECT is to uncover such spurious concepts and allow probing a black-box classifier. Motivated by real-world examples that revealed classifier dependency on surgical markings in identifying melanoma [86], we design this experiment. Given the synthetic nature of `SynthDerm` and how it has been designed based on real-world characteristics of melanoma [65, 80], each sample has a deterministic label of melanoma or benign. If the image is asymmetrical, has jagged borders, has different colors represented by salt-and-pepper noise, or has a large diameter (i.e., does not fit in a 40×40 square), the sample is melanoma. Otherwise, the image represents the benign class. Similar to in-situ dermatology images, melanoma samples have surgical markings more frequently than benign samples. We train a classifier to distinguish melanoma vs. benign lesion images.

Given a benign query, we would like to produce counterfactual explanations that depict *how* to modify the input sample to change its class membership. We want DISSECT to find CTs that disentangle meaningful characteristics of melanoma identification in terms of color, texture, or shape [65], and identify potential spurious artifacts that impact the classifier's predictions.

Tab. 2 summarizes the quantitative results on `SynthDerm`. Our method performs consistently well across all the metrics, significantly boosting *Distinctness* and *Substitutability* scores and making meaningful improvements on *Importance* scores. Our approach has higher performance compared

Figure 5: Examples from `CelebA`. A biased classifier has been trained to predict smile probability, where a training dataset was sub-sampled so that smiling co-occurs only with "bangs" and "blond hair" attributes. EPE does not support multiple CTs. EPE-mod converges on the same concept, despite having the ability to express various pathways to change $f(\bar{x})$ through $CT_1$ and $CT_2$. However, DISSECT discovers distinct pathways: $CT_1$ mainly changes hair color to blond, and $CT_2$ does not alter hair color but tries to add bangs.

Table 3: Quantitative results on `CelebA`. DISSECT performs better than or on par with the baselines in all categories. Notably, DISSECT greatly improves the *Distinctness* of CTs and achieves a higher *Realism* score, suggesting disentangling CTs does not diminish the quality of generated images and may even improve them. The classifier's precision, recall, accuracy when training on actual data is 95.4%, 98.6%, 92.7%, respectively.

| | Importance | | | | | | | Realism | | | | Distinctness | | | | | Substitutability | | | Stability | |
|---|---|---|---|---|---|---|---|---|---|---|---|---|---|---|---|---|---|---|---|---|---|
| | ↑R | ↑ρ | ↓KL | ↓MSE | ↑CF Acc | ↑CF Prec | ↑CF Rec | ↓FID | ↓Acc | ↓Prec | ↓Rec | ↑Acc | ↑Prec (micro) | ↑Prec (macro) | ↑Rec (micro) | ↑Rec (macro) | ↑Acc Sub | ↑Prec Sub | ↑Rec Sub | ↓CF MSE | ↓Prob JSD |
| CSVAE | 0.00 | 0.00 | 1.25 | 0.284 | 50.2 | 50.2 | 34.1 | 99.7 | 100 | 99.5 | 199.4 | 15.1 | 0.0 | 0.0 | 0.0 | 0.0 | 52.8 | 53.0 | **98.6** | 23.722 | 0.039 |
| EPE | 0.85 | **0.91** | 0.28 | 0.060 | 99.2 | **99.9** | 98.5 | 13.2 | 49.9 | 32.0 | 0.3 | - | - | - | - | - | **93.0** | 95.4 | 91.2 | **0.411** | **0.003** |
| EPE-mod | **0.86** | 0.90 | 0.21 | 0.048 | **99.5** | 99.7 | **99.2** | 13.3 | 49.3 | 33.3 | 0.1 | 18.7 | 50.0 | 50.1 | 15.2 | 15.2 | 91.8 | 94.8 | 89.4 | 0.446 | 0.004 |
| **DISSECT** | 0.84 | 0.88 | **0.19** | **0.047** | 99.2 | 99.8 | 98.5 | **11.1** | **49.2** | **0.0** | **0.0** | **95.0** | **98.0** | **98.1** | **96.1** | **96.1** | 91.9 | **96.9** | 87.6 | 0.567 | 0.005 |

to EPE-mod and EPE baselines and substantially improves upon CSVAE. Our method's high *Distinctness* and *Substitutability* scores show that DISSECT covers the landscape of potential concepts very well and retains the variety seen in real images strongly better than all the other baselines.

Fig. 4 illustrates a few examples to showcase DISSECT's improvements over the strongest baseline, EPE-mod. EPE-mod converges to finding a single concept that only vaguely represents meaningful ground-truth concepts. However, DISSECT successfully finds concepts describing asymmetrical shapes, jagged borders, and uneven colors that align with ABCDE of melanoma [65]. DISSECT also identifies surgical markings as a spurious concept impacting the classifier's decisions. Overall, the qualitative results show that DISSECT uncovers several critical blind spots of baselines.

### 4.5 CASE STUDY III: IDENTIFYING BIASES

Another potential use case of DISSECT is to identify biases that might need to be rectified. Since our approach does not depend on predefined user concepts, it may help discover unkwown biases. We design an experiment to test DISSECT in such a setting. We sub-sample `CelebA` to create a training dataset such that smiling correlates with "blond hair" and "bangs" attributes. In particular, positive samples either have blond hair or have bangs, and negative examples are all dark-haired and do not have bangs. We use this dataset to train a purposefully biased classifier. We employ DISSECT to generate two CTs. Fig. 5 shows the qualitative results, which depict that DISSECT automatically discovers the two biases, which other techniques fail to do. Tab. 3 summarizes the quantitative results that replicate our finding from Tab. 1 in § 4.3 and Tab. 2 in § 4.4 in a real-world dataset, confirming that DISSECT quantitatively outperforms all the other baselines in *Distinctness* without negatively impacting *Importance* or *Realism*.

## 5 CONCLUDING REMARKS

We present DISSECT that successfully finds multiple distinct concepts by generating a series of realistic-looking, counterfactually generated samples that gradually traverse a classifier's decision boundary. We validate DISSECT via quantitative and qualitative experiments, showing significant improvement over prior methods. Mirroring natural human reasoning for explainability [3, 4, 6, 83], our method provides additional checks-and-balances for practitioners to probe a trained model prior to deployment, especially in high-stakes tasks. DISSECT helps identify whether the model-under-test reflects practitioners' domain knowledge and discover biases not previously known to users.

One avenue for future explainability work is extending earlier theories [45, 71] that obtain disentanglement guarantees. While DISSECT uses an unsupervised approach for concepts discovery, it is possible to modify the training procedure and the concept disentangler network to allow for concept supervision, which can be another avenue for future work. Furthermore, extending this technique to domains beyond visual data such as time series or text-based input can broaden this work's impact. While we provide an extensive list of qualitative examples in the Appendix, human-subject studies to validate that CTs exhibit semantically meaningful attributes could further strengthen our findings.

## 6 ETHICS STATEMENT

We emphasize that our proposed method does not guarantee finding all the biases of a classifier, nor ensures semantic meaningfulness across all found concepts. Additionally, it should not replace procedures for promoting transparency in model evaluation such as model cards [51]; instead, we recommend it be used in tandem. We also acknowledge the limitations for Fitzpatrick ratings used in `SynthDerm` dataset in capturing variability, especially for darker skin tones [59, 77], and would like to consider other ratings for future.

## 7 REPRODUCIBILITY STATEMENT

The code for all the models and metrics is publicly available at `https://github.com/asmadotgh/dissect`.Additionally, we released the new dataset, which is publicly available at `https://affect.media.mit.edu/dissect/synthderm`. See §A.3.6 and §A.4 for additional details about evaluation metrics, experiment setup, and hyperparameter selection.

### ACKNOWLEDGMENTS

We thank Ardavan Saeedi, David Sontag, Zachary Lipton, Weiwei Pan, Finale Doshi-Velez, Emily Denton, Matthew Groh, Josh Belanich, James Wexler, Tom Murray, Kree Cole-Mclaughlin, Kimberly Wilber, Amirata Ghorbani, and Sonali Parbhoo for insightful discussions, Kayhan Batmanghelich and Sumedha Singla for the baseline open-source code, and MIT Media Lab Consortium for supporting this research.

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

# A  APPENDIX

## A.1  BASELINES

### A.1.1  BASELINE 1: MULTI-MODAL EXPLAINABILITY THROUGH VAE-BASED DISENTANGLEMENT

Disentanglement approaches have demonstrated practical success in learning representations that correspond to factors for variation in data [71], though some gaps between theory and practice remain [45]. However, the extent to which these techniques can aid post hoc explainability in conjunction with an external model is not well understood. Thus, we consider a set of baseline approaches based on VAEs explicitly designed for disentanglement: $\beta$-VAE [29], Annealed-VAE [8], $\beta$-TCVAE [11], and DIPVAE [41]. We extend each of them to incorporate the classifier's signal during their training processes for a fair comparison with DISSECT. Intuitively speaking, this encourages the generative model to learn latent dimensions that could influence the classifier, i.e., learning *Influential* CTs. Note that it is necessary to also cover data generative factors that are independent of the external classifier. This means that, along with having a good quality reconstruction and high likelihood, we need to promote sparsity in sensitivity of the external classifier to the latent dimensions of the generative model.

More formally, consider a vanilla VAE that has an encoder $e_\theta$ with parameters $\theta$, a decoder $d_\phi$ with parameters $\phi$, and the $M$-dimensional latent code $z$ with prior distribution $p(z)$. Recall that $x$ denotes the input sample. The objective of a VAE is to minimize the loss:

$$\mathcal{L}_{\theta,\phi}^{\text{vanilla VAE}} = -\mathbb{E}_{z \sim e_\theta(z|x)}[\log d_\phi(x|z)] + \mathbb{KL}(e_\theta(z|x)||p(z)).$$

We introduce an additional loss term for incorporating the black-box classifier's signal: $1/K \sum_{k=1}^{K} \partial f(x)/\partial z_k$. We impose this only for the first $K$ dimensions in the latent space[6],

---

[6]Without loss of generality, the additional term can be applied to the first $K$ dimensions, and there is no need to consider $\binom{K}{M}$ potential selections.

in other words, the number of desired CTs, $K \leq M$. Minimizing this term provides $CT_k$s, $k \in \{1, 2, \dots, K\}$, with negative $\partial f(x)/\partial z_k$, with a high $|\partial f(x)/\partial z_k|$. The final loss is:

$$\mathcal{L}_{\theta,\phi} = \mathcal{L}_{\theta,\phi}^{\text{vanilla VAE}} + \lambda * \frac{\sum_{k=1}^{K} \partial f(x)/\partial z_k}{K},$$

where $\lambda$ is a hyper-parameter. We apply this modification to the four aforementioned VAE-based approaches and refer to them with a -mod postfix, e.g. $\beta$-VAE-mod.[7]

**Development process.** To promote discovering *Important* CTs, we introduced $\mathcal{L}_{\text{aux}} = \frac{\sum_{d=1}^{K} \partial f(x)/\partial z_d}{K}$, which incorporated the directional derivative of $f$ with respect to the latent dimensions of interest into the loss function of VAE. Despite experimentation with many variants of $\mathcal{L}_{\text{aux}}$, we observed two common themes.

First, a monotonic increase of $f(\bar{x})$ through traversing one latent dimension and keeping the rest static was hardly achieved. Second, while the purpose of $\mathcal{L}_{\text{aux}}$ was to promote exerting *Importance* only in the first $K$ dimensions of the latent space, $\partial f(x)/\partial z_d$ for $d \in \{K+1, K+2, \cdots, M\}$ were impacted similarly. Having strongly correlated dimensions is a failure in achieving the very goal of disentanglement approaches. Table 4 summarizes a subset of the variants of $\mathcal{L}_{\text{aux}}$ studied.

Table 4: Summary of a subset of $\mathcal{L}_{\text{aux}}$ iterations. The development goal is to make the first $K$ dimensions of the latent space *Important*. In some iterations, we encouraged the remaining $M - K$ dimensions not to be *Important* to reduce potential correlation across latent dimensions.

| | $\mathcal{L}_{\text{aux}}$ |
|---|---|
| 1 | $\frac{\sum_{k=1}^{K} \partial f(x)/\partial z_k}{K}$ |
| 2 | $\frac{\sum_{k=1}^{K} \partial f(x)/\partial z_k}{K} + \frac{\sum_{d=K+1}^{M} |\partial f(x)/\partial z_d|}{M-K}$ |
| 3 | $\frac{\sum_{k=1}^{K} \partial f(x)/\partial z_k}{K} + \frac{\sum_{d=K+1}^{M} [\partial f(x)/\partial z_d]^2}{M-K}$ |
| 4 | $\frac{\sum_{d=K+1}^{M} |\partial f(x)/\partial z_d|}{M-K}$ |
| 5 | $\frac{\sum_{d=K+1}^{M} [\partial f(x)/\partial z_d]^2}{M-K}$ |
| 6 | $\frac{\sum_{k,d} |\partial f(x)/\partial z_d|/|\partial f(x)/\partial z_k|}{K*(M-K)}$ where $k \in \{1, 2, \cdots, K\}, d \in \{K+1, K+2, \cdots, M\}$ |
| 7 | $\frac{\sum_{k,d} \log(|\partial f(x)/\partial z_d|/|\partial f(x)/\partial z_k|)}{K*(M-K)}$ where $k \in \{1, 2, \cdots, K\}, d \in \{K+1, K+2, \cdots, M\}$ |

### A.1.2 BASELINE 2: MULTI-MODAL EXPLAINABILITY THROUGH CONDITIONAL SUBSPACE VAE

Another relevant area of work is conditional generation. In particular, Conditional subspace VAE (CSVAE) is a method aiming to solve unsupervised learning of features associated with a specific label using a low-dimensional latent subspace that can be independently manipulated [38]. CSVAE partitions the latent space into two parts: $w$ learns representations correlated with the label, and $z$ covers the remaining characteristics for data generation. An assumption of independence between $z$ and $w$ is made. To explicitly enforce independence in the learned model, we minimize the mutual information between $Y$ and $Z$. CSVAE has proven successful in providing counterfactual scenarios to reverse unfavorable decisions of an algorithm, also known as algorithmic recourse [17]. To adjust CSVAE to explain the decision-making of an external classifier $f$, we treat the predictions of the classifier as the label of interest.

More formally, the generative model can be summarized as:

$$w|y \sim N(\mu_y, \sigma_y^2.I), y \sim Bern(p),$$
$$x|w, z \sim N(d_{\phi_\mu}(w, z), \sigma_\epsilon^2.I), z \sim N(0, \sigma_z^2.I)$$

Conducting inference leads to the following objective function:

---

[7]We build these baselines on top of the open-sourced implementations provided in https://github.com/google-research/disentanglement_lib under Apache License 2.0.

$$M_1 = \mathbb{E}_{D(x,y)}[-\mathbb{E}_{q_\phi(z,w|x,y)}[\log p_\theta(x|w,z)] + \mathbb{KL}(q_\phi(w|x,y)||p_\gamma(w|y)) + \mathbb{KL}(q_\phi(z|x,y)||p(z)) - \log p(y)]$$

$$M_2 = \mathbb{E}_{q_\phi(z|x)}D(x)[\int_Y q_\delta(y|z)\log q_\delta(y|z)\,dy]$$

$$M_3 = \mathbb{E}_{q(z|x)D(x,y)}[q_\delta(y|z)]$$

$$\min_{\theta,\phi,\gamma} \beta_1 M_1 + \beta_2 M_2; \quad \max_\delta \beta_3 M_3$$

### A.1.3 BASELINE 3: MULTI-MODAL EXPLAINABILITY THROUGH PROGRESSIVE EXAGGERATION

Explanation by Progressive Exaggeration (**EPE**) [73] is a recent successful generative approach that learns to generate one series of counterfactual and realistic samples that change the prediction of $f$, given data and the classifier's signal. It is particularly relevant to our work as it explicitly optimizes *Influence* and *Realism*. EPE is a type of Generative Adversarial Network (GAN) [24] consisting of a discriminator ($D$) and a generator ($G$) that is based on Projection GAN [52]. It incorporates the amount of desired perturbation $\alpha$ on the outcome of $f$ as:

$$\mathcal{L}_{\text{cGAN}}(D) = -\mathbb{E}_{x \sim p_{data}}[\min(0, -1 + D(x, 0))] - \mathbb{E}_{x \sim p_{data}}[\min(0, -1 - D(G(x, \alpha), \alpha))] \quad (8)$$

$$\mathcal{L}_{\text{cGAN}}(G) = -\mathbb{E}_{x \sim p_{data}}[D(G(x, \alpha), \alpha)] \quad (9)$$

A Kullback–Leibler divergence (KL) term in the objective function between the desired perturbation ($\alpha$) and the achieved one ($f(G(x, \alpha))$) promotes Importance [73]:

$$\mathcal{L}_f(D, G) = r(D, G(x, \alpha)) + \mathbb{KL}(\alpha|f(G(x, \alpha))), \quad (10)$$

where the first term is the likelihood ratio defined in projection GAN [24], and [73] uses an ordinal-regression parameterization of it.

A reconstruction loss and a cycle loss promote self-consistency in the model, meaning that applying a reverse perturbation or no perturbation should reconstruct the query sample:

$$\mathcal{L}_{\text{rec}}(G) = ||x - G(x, f(x))||_1 \quad (11)$$

$$\mathcal{L}_{\text{cyc}}(G) = ||x - G(G(x, \alpha), f(x))||_1. \quad (12)$$

Thus, the overall objective function of EPE is the following:

$$\min_G \max_D \lambda_{\text{cGAN}}\mathcal{L}_{\text{cGAN}}(D, G) + \lambda_f \mathcal{L}_f(D, G) + \lambda_{\text{rec}}\mathcal{L}_{\text{rec}}(G) + \lambda_{\text{rec}}\mathcal{L}_{\text{cyc}}(G), \quad (13)$$

where $\lambda_{\text{cGAN}}$, $\lambda_f$, and $\lambda_{\text{rec}}$ are the hyper-parameters.

Note that EPE only finds one pathway to switch the classifier's outcome. We argue that classifiers learned from challenging and realistic datasets will have complex reasoning pathways that could enhance model explainability if revealed. Decomposing this complexity is needed to make reasoning comprehensible for humans. We thus create a more powerful baseline, an EPE-variant, **EPE-mod**. EPE-mod learns multiple pathways by making the generator conditional on another variable: the CT dimension. More formally, EPE-mod updates $G(\cdot, \cdot)$ to $G(\cdot, \cdot, k)$ in Eq. equation 8-equation 12, while Eq. equation 13 remains unchanged. We compare DISSECT to both EPE and EPE-mod.

### A.2 DISSECT DETAILS

We build our proposed method on EPE-mod and further promote distinctness across CTs by adding a disentangler network, $R$. The disentangler is a classifier with K classes. Given a pair of $(x, x')$ images, $R$ tries to predict which $\text{CT}_{k;k\in\{1,...,K\}}$ has perturbed query $x$ to produce $x'$. Note that $R$ can return close to 0 probability for all classes if $x'$ is just a reconstruction of $x$, indicating no tweaked dimensions. The disentangler also penalizes the generator if any CTs use similar pathways to cross the decision boundary. See the appendix for schematics of our method.

To formalize this, let:

$$\hat{x}_k = G(x, f(x), k)$$

$$\bar{x}_k = G(x, \alpha, k)$$

$$\tilde{x}_k = G(\bar{x}_k, f(x), k).$$

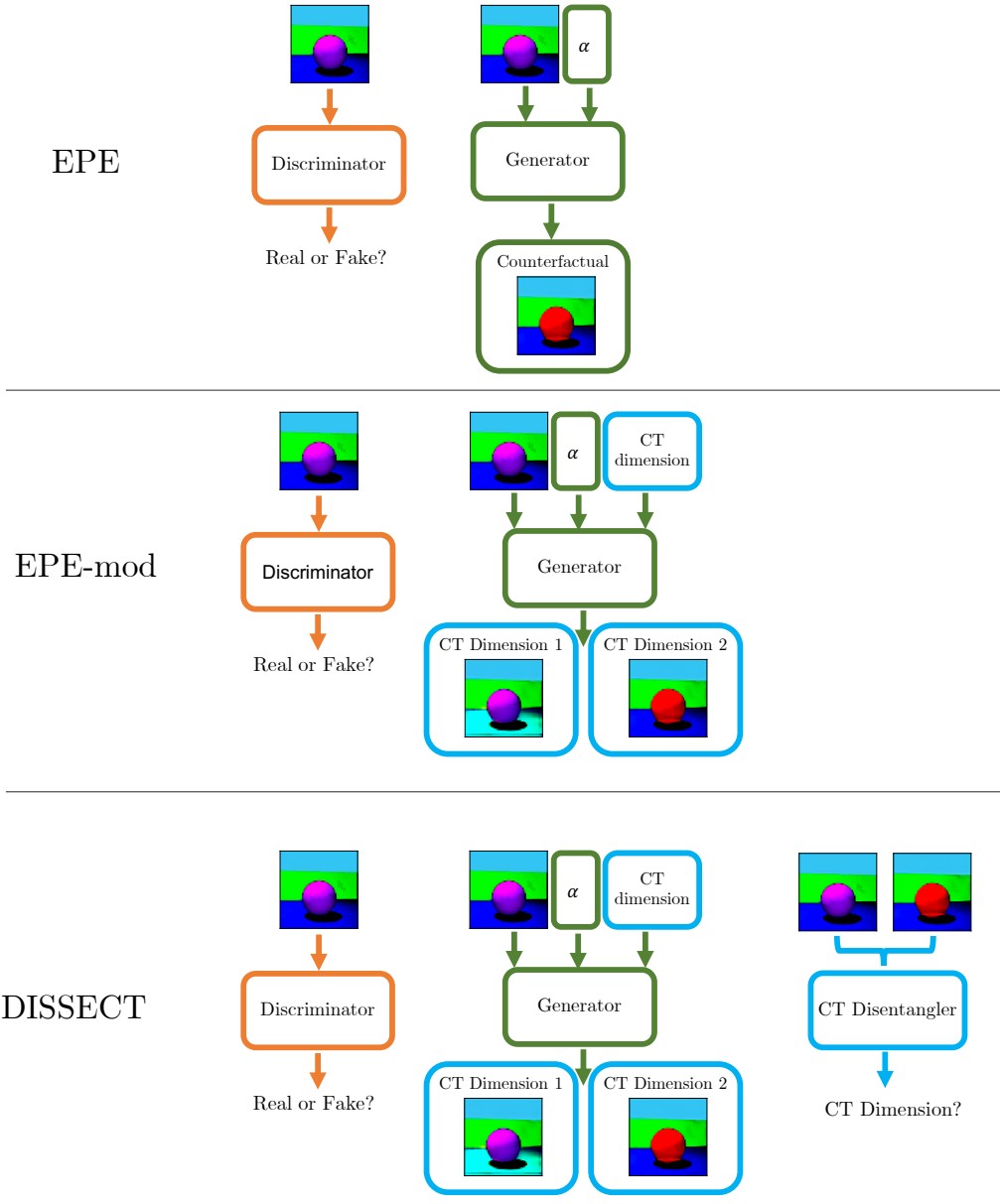

Figure 6: Simplified illustration of EPE, EPE-mod, and DISSECT. EPE-mod can be viewed as an ablated version of DISSECT. Orange, Green, and Blue show elements related to the discriminator, generator, and CT disentangler, respectively. The novelty of DISSECT is the disentanglement components.

Note that $\hat{x}_k$ and $\tilde{x}_k$ are reconstructions of $x$ while $\bar{x}_k$ is perturbed to change the classifier output from $f(x)$ to $\alpha$. Therefore, $x$, $\bar{x}_k$ and $\tilde{x}_k$ form a cycle, and $k$ represents $CT_k$. $R(.,.)$ is the predicted probabilities of the perturbed concept given a pair of examples, which is a vector of size $K$, where each element is a value in $[0,1]$. We define the following cross entropy loss that is a function of both $R$ and $G$:

$$\mathcal{L}_r(G,R) = CE(e_k, R(x, \bar{x}_k)) + CE(e_k, R(\bar{x}_k, \tilde{x}_k))$$

$$= -\mathbb{E}_{x \sim p_{data}} \sum_{k=1}^{K} [e_k log R(x, \bar{x}_k) + e_k log R(\bar{x}_k, \tilde{x}_k)] \quad (14)$$

Here, $e_k$ refers to a one-hot vector of size $K$ where the $k$-th element is one and the remaining elements are zero. This term enforces $R$ to identify no change when receiving reconstructions of the same image as input and utilizes the cycle and promotes determining the correct dimension when a non-zero change has happened, either increasing or decreasing the outcome of $f$. In summary, the overall objective function of our method is:

$$\min_{G,R} \max_D [\lambda_{\text{cGAN}} \mathcal{L}_{\text{cGAN}}(D, G) + \lambda_f \mathcal{L}_f(D, G) + \lambda_{\text{rec}} \mathcal{L}_{\text{rec}}(G) + \lambda_{\text{rec}} \mathcal{L}_{\text{cyc}}(G) \\ + \lambda_r \mathcal{L}_r(G, R)] \tag{15}$$

For this adversarial min-max optimization, we use the Adam optimizer [37]. See Figure 6 for a for a visualization of DISSECT .We open-source our implementation at `https://github.com/asmadotgh/dissect` under MIT license, which builds on top of the open-source implementation of EPE [73] [8].

## A.3 EVALUATION METRICS DETAILS

In this section, we provide more details about the evaluation metrics. First, we provide evidence supporting these criteria as desirable qualities for explainability. Then, we summarize how these qualities have been measured in prior work. Finally, we explain when we can use prior formulation of these metrics as is, and when we need to modify them to make them applicable to the counterfactual explanation case. We justify how these formulations capture the desired criteria.

### A.3.1 IMPORTANCE

**Motivation.** Explanations should produce the desired outcome from the black-box classifier $f$. Previous work has referred to this quality using different terms, such as importance [22], compatibility with classifier [73], and classification model consistency [74].

**Quantification.** Most previous methods have relied on visual inspection. Singla et al. [74] have plotted expected outcome from the classifier, against the actual classifier prediction of the generated explanations. Ghorbani et al. [22] have plotted prediction accuracy vs. adding or removal of discovered concepts.

We introduce a quantitative metric to measure the gradual increase of the target class's posterior probability through a CT. Notably, we compute the correlation between $\alpha$ and $f(I(x, \alpha, k; f))$ introduced in § 3. For brevity, we refer to $f(I(x, \alpha, k; f))$ as $f(\bar{x})$ in the remainder of the paper. We also report the mean-squared error and the Kullback–Leibler (KL) divergence between $\alpha$ and $f(\bar{x})$. This is one way to summarize information in the plots used by Singla et al. [74] for *Importance* evaluation.

We also calculate the performance of $f$ on the counterfactual explanations. Specifically, we replace the test set of real images with their generated counterfactual explanations and quantify the performance of the pre-trained black-box classifier $f$ on the counterfactual test set. Better performance suggests that counterfactual samples are compatible with $f$ and lie on the correct side of the classifier's boundary. This is one way to summarize the information in the plots used by Ghorbani et al. [22] for *Importance* evaluation.

### A.3.2 REALISM

**Motivation.** We need the generated samples that form a CT to look realistic to enable users to identify concepts they represent. It means the counterfactual explanations should lie on the data manifold. This quality has been referred to as realism or data consistency [73].

**Quantification.** A variety of inception scores have perviously been used to quantify visual quality of generated images, such as Fréchet Inception Distance [28] and Inception Score [67].

---

[8] `https://github.com/batmanlab/Explanation_by_Progressive_Exaggeration` available under MIT License.

Similar to prior work, we use FID [28]. However, since FID is based on an Inception V3 network trained on ImageNet, it could be a better fit for more realistic data that has similar distribution to ImageNet. This is not the case, especially for `3d Shapes` and `SynthDerm` in our work. Therefore, we include an additional metric inspired by [18]. We train a post hoc classifier that predicts whether a sample is real or generated. We report performance of this classifier as a proxy for *Realism*.

### A.3.3 DISTINCTNESS

**Motivation.** Another desirable quality for explanations is to represent inputs with non-overlapping concepts, often referred to as diversity [50]. Others have suggested similar properties such as coherency, meaning examples of a concept should be similar but different from examples of other concepts [22].

**Quantification.** To quantify this in a counterfactual generation setup, we introduce a distinctness measure capturing the performance of a secondary classifier that distinguishes between CTs. We train a classifier post hoc that given a query image $x$ and a generated image $x'$ and $K$ number of CTs, predicts $CT_k$ that has perturbed $x$ to produce $x'$, $k \in \{1, \ldots, K\}$. This

### A.3.4 SUBSTITUTABILITY

**Motivation.** The representation of a sample in terms of concepts should preserve relevant information [50, 60]. Previous work has formalized this quality for counterfactual generation contexts through a proxy called substitutability [68].

**Quantification.** Substitutability measures an external classifier's performance on real data when it is trained using only synthetic images. This metric has been used in other contexts outside of explainability and has been called Classification Accuracy Score (CAS) [62]. CAS is more broadly applicable than Fréchet Inception Distance [28] & Inception Score [67] that are only useful for GAN evaluation. Furthermore, CAS can reveal information that none of these inception scores successfully capture [62]. A higher substitutability score suggests that explanations have retained relevant information and are of high quality.

We follow Samangouei et al. [68] for formulation of this metric.

### A.3.5 STABILITY

**Motivation.** Explanations should be coherent for similar inputs, a quality known as stability [21, 50, 60].

**Quantification.** To quantify stability in counterfactual explanations, we augment the test set with additive random noise to each sample $x$ and produce $S$ randomly augmented copies $x_i^{''}$. Then, we generate counterfactual explanations $\bar{x}$ and $\bar{x}_i^{''}$, respectively. We calculate the mean-squared difference between counterfactual images $\bar{x}$ and $\bar{x}_i^{''}$ and the resulting Jensen Shannon distance (JSD) between the predicted probabilities $f(\bar{x})$ and $f(\bar{x}_i^{''})$.

### A.3.6 IMPLEMENTATION DETAILS

The VAE-based baselines support continuous values for latent dimensions $z_k$. Also, we can directly sample latent code values and produce $CT_k$ by keeping $z_j$ ($j \neq k$) constant and monotonically increasing $z_k$ values. However, to calculate the evaluation metrics comparably to EPE, EPE-mod, and DISSECT, we do the following: We encode each query sample using the probabilistic encoder. We set $z_j = \mu_j$, $j \neq k$ where $\mu_j$ is the mean of the fitted Gaussian distribution for $z_j$. For dimension $k$, we produce $N + 1$ linearly spaced values between $\mu_p \pm 2 * \sigma_p$, where $\mu_p$ and $\sigma_p$ are the mean and standard deviation of the prior normal distribution, in our case 0.0 and 1.0 respectively. Note that these different values for $z_k$ map out to $\alpha$, $\alpha \in \{0, \frac{1}{N}, \cdots, 1\}$ in EPE, EPE-mod, and DISSECT models. After this step, calculating all the metrics related to *Importance*, *Realism*, and *Distinctness* is identical across all the models.

Table 5: Summary of hyper-parameter values. Discriminator optimization happens once every $D$ steps. Similarly, generator optimization happens once every $G$ steps. $\lambda_r$ is specific to DISSECT, and $K$ is specific to EPE-mod and DISSECT. All the remaining parameters are shared across EPE, EPE-mod, and DISSECT. Note that samples used for evaluation are not included in the training process.

| | Preprocessing | | Training | | | | | | | | | Evaluation Metrics | | | |
|---|---|---|---|---|---|---|---|---|---|---|---|---|---|---|---|
| | $N$ | max samples per bin | $\lambda_{cGAN}$ | $\lambda_{rec}$ | $\lambda_f$ | $D$ steps | $G$ steps | batch size | epochs | $K$ | $\lambda_r$ | max # samples | batch size | epochs | hold-out test ratio |
| 3D Shapes | 3 | 5,000 | 1 | 100 | 1 | 1 | 5 | 32 | 300 | 2 | 10 | 10,000 | 32 | 10 | 0.25 |
| SynthDerm | 2 | 1,350 | 2 | 100 | 1 | 5 | 1 | 32 | 300 | 5 | 2 | 10,000 | 8 | 10 | 0.25 |
| CelebA | 10 | 5,000 | 1 | 100 | 1 | 1 | 5 | 32 | 300 | 2 | 2 | 10,000 | 32 | 10 | 0.25 |

## A.4 EXPERIMENT SETUP AND HYPER-PARAMETER TUNING DETAILS

Experiments were conducted on an internal compute cluster at authors' institution. Training and evaluation of all models across the three datasets have approximately taken 1000 hours on a combination of Nvidia GPUs including GTX TITAN X, GTX 1080 Ti, RTX 2080 Rev, and Quadro K5200.

We seeded the model's parameters from [73] based on the reported values in their accompanying open-sourced repository.[9] We used the same parameters for 3D Shapes, except for the number of bins, $N$, used for ordinal regression transformation of the classifier's posterior probability. The largest number of bins that resulted in non-zero samples per bin, 3, was selected. We kept all the parameters shared between EPE, EPE-mod, and DISSECT the same.

Given the experiments' design, we fixed the number of dimensions $K$ in DISSECT and EPE-mod to 2. We experimented with a few values for $\lambda_r$, 1, 10, 20, 50. Based on manual inspection after 30k training batches, $\lambda_r$ was selected. Factors considered for selection included inspecting the perceived quality of generated samples and the learning curves of $\mathcal{L}_{cGAN}(D)$, $\mathcal{L}_{cGAN}(G)$, $\mathcal{L}_{cyc}(G)$, $\mathcal{L}_{rec}(G)$, and $\mathcal{L}_r(G, R)$.

For evaluation, we used a hold-out set including 10K samples. For post hoc evaluation classifiers predicting *Distinctness* and *Realism*, 75% of the samples were used for training, and the results were reported on the remaining 25%. See Table 5 for the summary of the hyper-parameter values.

## A.5 ADDITIONAL QUALITATIVE RESULTS

### A.5.1 CASE STUDY I

Recall that considering 3D Shapes, we define an image as "colored correctly" if the shape hue is red *or* the floor hue is cyan. Given a not "colored correctly" query, we recover a CT related to the shape color and another CT associated with the floor color–two different pathways leading to switching the classifier outcome for that sample. See Figure 7 for additional qualitative examples where classification outcome is flipped from False to True.

However, these two ground-truth concepts do not directly apply to switching the classifier outcome from True to False in this scenario. For example, if an image has a red shape *and* a cyan floor, both colors need to be changed to switch the classification outcome. As shown in Figure 8, we still observe that applying DISSECT to such cases results in two discovered CTs that change different combinations of colors while EPE-mod converges to the same CT.

### A.5.2 CASE STUDY III

Recall the biased CelebA experiment where smiling correlates with "blond hair" and "bangs" attributes. Figure 9 shows additional qualitative samples, suggesting that DISSECT can recover and separate the aforementioned concepts, which other techniques fail to do.

---

[9]https://github.com/batmanlab/Explanation_by_Progressive_Exaggeration available under MIT License.

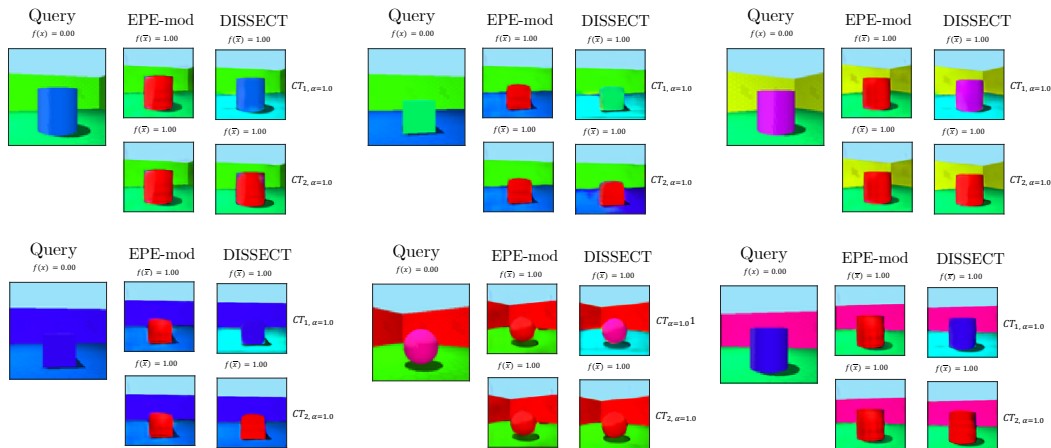

Figure 7: Qualitative results on `3D Shapes` when flipping classification outcome from "False" to "True." We observe that EPE-mod converges to finding the same concept, despite having the ability to express multiple pathways to switch the classifier outcome. However, DISSECT can discover the two Distinct ground-truth concepts: $CT_1$ flips the floor color to cyan, and $CT_2$ converts the shape color to red.

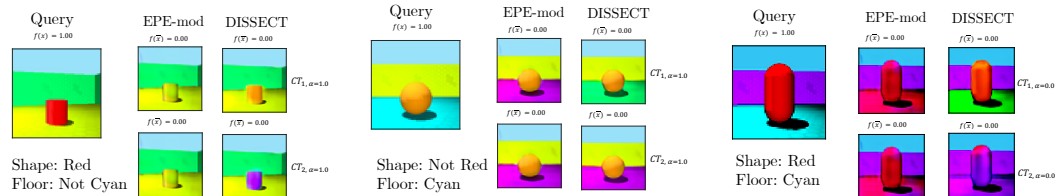

Figure 8: Qualitative results on `3D Shapes` when flipping classification outcome from "True" to "False." We observe that EPE-mod converges to finding the same concept, despite having the ability to express multiple pathways to switch the classifier outcome. However, DISSECT is capable of discovering Distinct paths to do so. **Left**: When the input query has a red shape, but the floor color is not cyan, $CT_1$ flips the shape color to orange and $CT_2$ flips it to violet. **Middle**: When the input query has a cyan floor, but the shape color is not red, $CT_1$ flips the floor color to lime, and $CT_2$ converts it to magenta. **Right**: When the input query has a red shape and cyan floor, $CT_1$ changes the shape color to dark orange and floor color to lime, and $CT_2$ flips the shape color to violet and floor color to magenta.

### A.5.3  LIMITATIONS

While we provide several qualitative examples, further confirmation from human-subject studies to validate that CTs exhibit semantically meaningful attributes could strengthen our findings.

The number of concepts to be discovered by DISSECT is a hyper parameter that can be selected by the user. While *Distinctness* and *Substitutability* are designed to be globally evaluated across all concepts, *Importance*, *Realism*, and *Stability* scores can be calculated for each concept. We hypothesize that ranking the discovered concepts based on these individual scores can benefit the users. In particular, it can help them focus on more salient concepts and not be overwhelmed by less informative concepts. Future user-studies are required to investigate these hypotheses.

We emphasize that our proposed method does not guarantee finding all the biases of a classifier, nor ensures semantic meaningfulness across all found concepts. One avenue for future explainability work is extending earlier theories [45, 71] that obtain disentanglement guarantees.

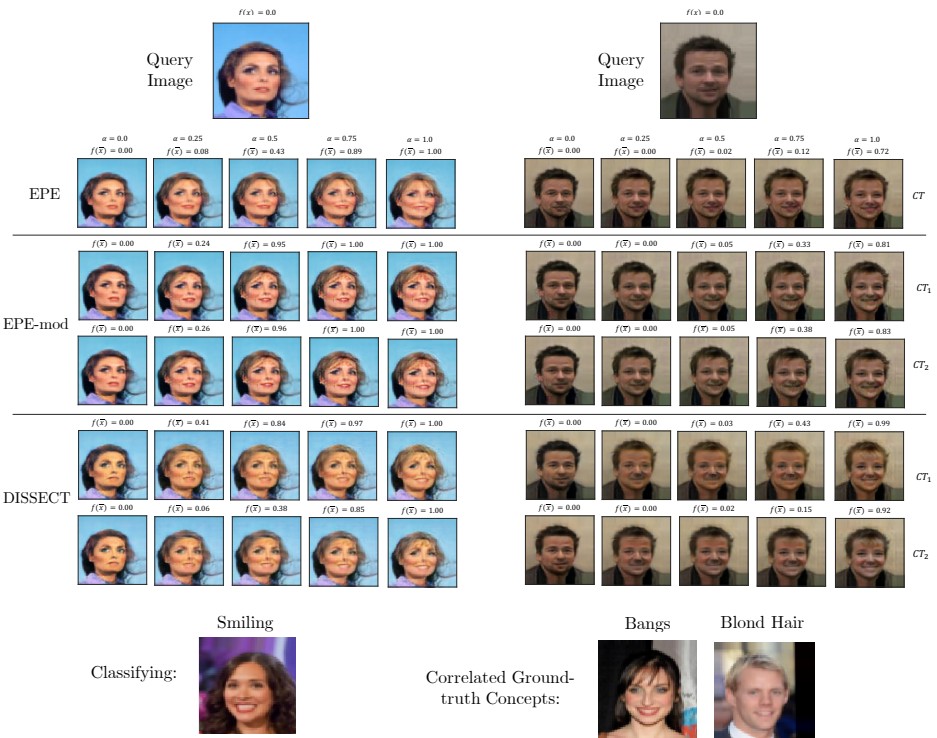

Figure 9: Qualitative results on `CelebA`. A biased classifier has been trained to predict smile probability, where the training dataset has been sub-sampled such that smiling co-occurs only with "bangs" and "blond hair" attributes. EPE does not support multiple CTs. We observe that EPE-mod converges to finding the same concept, despite having the ability to express several pathways to change $f(\bar{x})$ through $CT_1$ and $CT_2$. However, DISSECT can discover Distinct routes: $CT_1$ mainly changes hair color to blond, and $CT_2$ does not alter hair color but focuses more on hairstyle and tries to add bangs. Thus it identifies two otherwise hidden biases.

Table 6: Ablation experiment. Different rows show results for `CelebA`, with different $\lambda_r$ values. All the other variables are the same across experiments: $\lambda_{cGAN} = 1$, $\lambda_{rec} = 100$, $\lambda_f = 1$.

| | $\lambda_r$ | Importance | | | | | | Realism | | | Distinctness | | | | | Substitutability | | | Stability | |
| | | ↑R | ↑$\rho$ | ↓KL | ↓MSE | ↑CF Acc | ↑CF Prec | ↑CF Rec | ↓Acc | ↓Prec | ↓Rec | ↑Acc | ↑Prec (micro) | ↑Prec (macro) | ↑Rec (micro) | ↑Rec (macro) | ↑Acc Sub | ↑Prec Sub | ↑Rec Sub | ↓CF MSE | ↓Prob JSD |
|---|---|---|---|---|---|---|---|---|---|---|---|---|---|---|---|---|---|---|---|---|---|
| EPE-mod | 0 | 0.859 | 0.904 | 0.212 | 0.048 | 99.465 | 99.749 | 99.180 | 49.26 | 33.333 | 0.118 | 18.704 | 50.008 | 50.108 | 15.217 | 15.237 | 91.810 | 94.824 | 89.434 | 0.446 | 0.004 |
| DISSECT | 0.01 | 0.813 | 0.868 | 0.329 | 0.067 | 98.4 | 99.8 | 97.0 | 49.8 | 0.0 | 0.0 | 36.1 | 53.6 | 54.5 | 47.3 | 47.4 | 98.4 | 95.1 | 85.2 | 0.796 | 0.004 |
| DISSECT | 0.1 | 0.866 | 0.915 | 0.246 | 0.046 | 99.448 | 99.829 | 99.065 | 49.64 | 36.364 | 0.799 | 82.304 | 93.145 | 93.210 | 82.385 | 82.394 | 81.122 | 97.291 | 66.236 | 0.453 | 0.003 |
| DISSECT | 1 | 0.821 | 0.877 | 0.309 | 0.060 | 98.640 | 99.785 | 97.490 | 49.480 | 0.000 | 0.000 | 96.024 | 97.594 | 97.595 | 97.730 | 97.732 | 90.524 | 92.367 | 89.524 | 0.613 | 0.004 |
| **DISSECT** | **2** | 0.843 | 0.882 | 0.188 | 0.047 | 99.165 | 99.843 | 98.485 | 49.16 | 0.0 | 0.0 | 94.980 | 98.042 | 98.108 | 96.056 | 96.053 | 91.938 | 96.937 | 87.559 | 0.567 | 0.005 |
| DISSECT | 10 | 0.044 | 0.110 | 2.528 | 0.324 | 52.122 | 52.174 | 50.945 | 98.32 | 98.917 | 97.740 | 96.012 | 98.129 | 98.132 | 97.162 | 97.163 | 43.033 | 29.333 | 5.293 | 0.414 | 0.002 |
| DISSECT | 100 | 0.207 | 0.144 | 1.416 | 0.270 | 59.76 | 83.096 | 24.505 | 99.68 | 99.841 | 99.524 | 96.328 | 97.582 | 97.582 | 98.170 | 98.170 | 54.863 | 63.840 | 34.266 | 0.180 | 0.001 |

## A.6 ABLATION/SENSITIVITY EXPERIMENTS

EPE-mod can be viewed as the ablated version of DISSECT where $\lambda_r = 0$. For a more detailed analyses of the influence of the disentanglement element, we conducted an experiment with different $\lambda_r$ while fixing the other hyperparameters of the model. As seen in Table 6, increasing $\lambda_r$ up to a point improves *Distinctness* and retains performance on the other criteria of interest. Further increasing $\lambda_r$ hurts the quality of generated images, and thus negatively impacts *Realism* and *Substitutability*.

## A.7 INDIVIDUAL CONCEPT INFLUENCE

Note that metrics such as *Distinctness* and *Substitutability* are not meaningful for each concept. However, *Importance*, *Realism*, and *Stability* can be calculated on each concept separately, or over all concepts. Calculating such metrics for each concept can help us rank them with respect to that criterion. For example, *Importance* scores per concept can tell us which concept is more important

Table 7: Individual *Importance* scores per concept discovered by DISSECT. Discovered `3D Shapes` concepts: 1) Red shape, 2) Cyan floor. Discovered `SynthDerm` concepts: 1) Color, 2) Diameter, 3) Asymmetry, 4) Border, 5) Surgical markings and color. Discovered `CelebA` concepts: 1) Blond hair, 2) Bangs.

| | | | | | | Importance | | |
|---|---|---|---|---|---|---|---|---|
| | | $\uparrow$ R | $\uparrow \rho$ | $\downarrow$ KL | $\downarrow$ MSE | $\uparrow$ CF Acc | $\uparrow$ CF Prec | $\uparrow$ CF Rec |
| `3D Shapes` | $CT_1$ | 0.823 | 0.843 | 1.421 | 0.084 | 92.32 | 100.0 | 98.64 |
| | $CT_2$ | 0.822 | 0.766 | 1.794 | 0.085 | 98.15 | 100.0 | 96.3 |
| `SynthDerm` | $CT_1$ | 0.908 | 0.747 | 0.383 | 0.021 | 97.615 | 91.886 | 89.671 |
| | $CT_2$ | 0.895 | 0.746 | 0.423 | 0.023 | 97.215 | 90.191 | 88.294 |
| | $CT_3$ | 0.935 | 0.75 | 0.276 | 0.015 | 98.26 | 93.61 | 93.037 |
| | $CT_4$ | 0.929 | 0.749 | 0.303 | 0.016 | 98.195 | 93.244 | 92.923 |
| | $CT_5$ | 0.919 | 0.748 | 0.349 | 0.018 | 97.935 | 92.605 | 91.507 |
| `CelebA` | $CT_1$ | 0.845 | 0.882 | 0.184 | 0.046 | 99.08 | 99.817 | 98.34 |
| | $CT_2$ | 0.842 | 0.882 | 0.192 | 0.047 | 99.25 | 99.868 | 98.63 |

Table 8: Interaction between generation quality and explanation quality. Three different sizes have been considered for the underlying GAN architecture: `xxsmall`, `small`, `base`. For comparison, a `GAN-only` version of `base` is also included, where all the other components are disabled, i.e. $\lambda_r = 0$, $\lambda_{rec} = 0$, and $\lambda_f = 0$.

| | Importance | | | | Realism | | | | Distinctness | | | | |
|---|---|---|---|---|---|---|---|---|---|---|---|---|---|
| | $\uparrow$ R | $\uparrow \rho$ | $\downarrow$ KL | $\downarrow$ MSE | $\downarrow$ FID | $\downarrow$ Acc | $\downarrow$ Prec | $\downarrow$ Rec | $\uparrow$ Acc | $\uparrow$ Prec (micro) | $\uparrow$ Prec (macro) | $\uparrow$ Rec (micro) | $\uparrow$ Rec (macro) |
| `GAN-only` | 0.000 | 0.000 | 6.049 | 0.412 | 7.367 | 49.32 | 0.0 | 0.0 | 33.287 | 0.0 | 0.0 | 0.0 | 0.0 |
| `xxsmall` | 0.0 | 0.0 | 11.526 | 0.414 | 110.964 | 100.0 | 100.0 | 100.0 | 33.3 | 0.0 | 0.0 | 0.0 | 0.0 |
| `small` | 0.822 | 0.803 | 2.915 | 0.085 | 51.002 | 99.08 | 99.15 | 98.99 | 99.980 | 99.990 | 99.990 | 99.980 | 99.980 |
| `base` | 0.84 | 0.88 | 0.19 | 0.047 | 42.109 | 49.2 | 0.0 | 0.0 | 95.0 | 98.0 | 98.1 | 96.1 | 96.1 |

in the decision making of the classifier. Table 7 summarizes the *Importance* scores per concept. As expected, given the symmetric design of our experiments, all discovered concepts exhibit similar *Importance* scores.

## A.8 INTERACTION BETWEEN GENERATION QUALITY AND EXPLANATION QUALITY

In this section, we evaluate DISSECT with different sizes of underlying GAN architectures. As shown in Table 8, even with imperfect generation quality, DISSECT can still discover relevant concepts with comparable performance in terms of *Importance*, *Distinctness*. This is despite significant impacts on the *Realism* scores. Perfect generation quality can further improve realism. However, a reasonable generation quality is sufficient for successful explanation with DISSECT.

With the recent advances in generative modeling that have achieved great generation quality over a majority of domains, it is timely to utilize them when available for generating realistic-looking explanations. It is worth mentioning that a perfect generation alone is not sufficient for good explanations in terms of performance criteria other than *Realism*. For comparison, we have calculated performance scores of a strong generative model, excluding the interpretability-related terms in the objective function. See the results in Table 8 for the `GAN-only` row.

