# OpenReview forum: "DISSECT: Disentangled Simultaneous Explanations via Concept Traversals"
_ICLR.cc/2022/Conference — ICLR 2022 Poster_

### Official Review · Reviewer_47RT · 2021-10-28

**Correctness:** 3
**Technical Novelty And Significance:** 2
**Empirical Novelty And Significance:** 3
**Recommendation:** 6
**Confidence:** 4

**Main Review:**

This is an original paper that introduces a novel approach to model explanations. It builds on several advancements in the literature, and aims at producing a solution that is useful for the users that will actually run it. It is clearly written and easy to follow, and includes an extensive experimental analysis that compares the approach to many strong baselines and on several substantially different dataset.

While I do think that this is overall a good paper, I do have some minor issues that I would like to see clarified. First, I am worried about the dependency on a strong generative model that can successfully disentangle concepts. It could be that this approach would fail on data with structures that do not allow such an easy generation process, such as natural language. To strengthen the findings in the paper, I would like to see some exploration of the relationship between generation quality and explanation quality.

Second, I’m not sure that the fact that this method does not depend on predefined user concepts is a good thing. It can certainly be advantageous if we want to explore hidden biases, but it could also be problematic in cases where we want to test a specific hypothesis regarding a concept we already know and care about. How would an approach like DISSECT deal with cases where we care about some specific concepts, but would also like to discover unknown biases?

Thirdly, I think that the paper might benefit from a deeper discussion of the importance of generating realistic counterfactuals from a causal perspective. The main motivation for producing counterfactuals for explanations is the estimation of causal effects, and connecting DISSECT to this literature can improve the paper in my opinion.

Fourthly, this paper seems fairly based on a specific paper, “Explanation by Progressive Exaggeration”. While the authors are clear about their use of this work, the similarity of the approaches is significant. In fact, the overall objective is identical except for the extra term that enforces disentanglement. Given that, the technical contribution of this paper is a bit thin.

Lastly, I have trouble understanding the qualitative result in Figure 5. From carefully observing it, I am not convinced that DISSECT does a substantially better job than EPE-mod in disentangling the confounding concepts. While I do find the quantitative evidence convincing, I would consider clarifying or replacing Figure 5.


**Summary Of The Paper:**

This paper proposes a generative approach to model explanations, where explanations are composed of automatically generated counterfactual examples that differ by the observed magnitude of a given learned concept. Through a series of extensive experiments, the authors show that the proposed approach, DISSECT, can successfully disentangle useful and realistic concepts.

The main contributions of the paper is the novel approach that generates several counterfactual examples across a concept spectrum, and a new synthetic dermatology dataset that allows testing such methods where they matter most.


**Summary Of The Review:**

While I do think that this is overall a good paper, I do have some minor issues that I would like to see clarified. First, I am worried about the dependency on a strong generative model that can successfully disentangle concepts. Second, I’m not sure that the fact that this method does not depend on predefined user concepts is a good thing. Thirdly, I think that the paper might benefit from a deeper discussion of the importance of generating realistic counterfactuals from a causal perspective. Lastly, I think that the technical contribution might be a bit too thin, given prior work that this paper builds on.

---

> ### Author Response · Authors · 2021-11-23
> **Author response**
>
> We thank the reviewer for their comprehensive and insightful feedback.
>
> ## Depending on a strong generative model:
>
> **We have added a new section to the appendix, A.8, discussing the relationship between generation quality and explanation quality.**
>
> We conduct a new experiment, evaluating DISSECT with different sizes of underlying GAN architectures. Our results show that even with imperfect generation quality, DISSECT can still discover relevant concepts with comparable performance in terms of importance and distinctness. This is despite significant impacts on the realism scores. Perfect generation quality can further improve realism. However, a reasonable generation quality is sufficient for successful explanation with DISSECT. **Table 8** summarizes the results of this experiment.
>
> With the recent advances in generative modeling that have achieved great generation quality over a majority of domains, it is timely to utilize them when available for generating realistic-looking explanations. It is worth mentioning that a perfect generation alone is not sufficient for good explanations in terms of performance criteria other than realism. For comparison, we have calculated performance scores of a strong generative model, excluding the interpretability-related terms in the objective function. See the results in **Table 8** for the GAN-only row.
>
>
> ## Depending on predefined user concepts:
> We agree that quantifying the dependence of a model on a user-defined concept is an important area. Prior approaches exist that require supervision for the concepts, such as [9, 35]. Such approaches successfully equip users to test hypotheses about their concept of interest and its relevance to a classifier. However, this is outside the scope of this paper. The goal of our DISSECT work is to address the **unsupervised** discovery and **disentanglement** of concepts. DISSECT can also be used in tandem with such approaches to support both supervised and unsupervised concepts.
>
> ## EPE and DISSECT:
> Since the focus of our approach is highlighting the importance of the disentanglement piece and thorough evaluation and experimentation, we have tried to leverage previous work wherever possible. Due to the strong performance of EPE as an interpretability baseline that also has high generation quality, we follow EPE for building DISSECT’s backbone. We set out to reuse EPE elements, and limit the new added variables and modifications only when necessary for supporting disentanglement. We would like to borrow quotes from reviewer WoTB and reviewer 1keG to highlight the significance of disentangling multiple counterfactual explanations: WoTB:  “to obtain multiple counterfactuals is novel and quite useful for multiple practical applications.” 1keG: “[this paper] creates several explanations which increasingly rely on a concept. I think this aspect is very interesting and could be relevant in practice. For example, as highlighted in the paper, it can be used to depict how a mole turning from benign to malignant may progress.” **We have elaborated on these points in Appendix A.2 for further clarification.**
>
> ## Figure 5 clarification:
> We have **replaced Fig. 5** with a clearer example.

---

### Official Review · Reviewer_1keG · 2021-10-30

**Correctness:** 3
**Technical Novelty And Significance:** 3
**Empirical Novelty And Significance:** 3
**Recommendation:** 6
**Confidence:** 3

**Main Review:**

**Strengths**:

- novelty: the approach taken in this paper is very different from other counterfactual explanation methods. This is the first paper I've read that creates several explanations which increasingly rely on a concept. I think this aspect is very interesting and could be relevant in practice. For example, as highlighted in the paper, it can be used to depict how a mole turning from benign to malignant may progress.
- experiments/evaluation: The authors demonstrate the efficacy of their method over three different datasets (each well motivated for the problem at hand) and extensively evaluate the performances. Specifically, they introduce metrics for each of the desiderata  ("importance", "realism", "distinctness", "stability").  The proposed method generally performs better than the benchmarks (VAEs designed for disentanglement, CSVAE, EPE, and modifications thereof).

**Weaknesses**:

- several of the evaluation metrics are first introduced in this paper. It would be great to see some more justification on whether the evaluation metrics actually measure what they are trying to target. E.g., for realism, one could simulate data for which we know the "ground truth" and then evaluate the metric.
- to a certain extent, the method is designed to target these metrics. As such, I would expect it to perform better. Having said that, I think this is only a small weakness. Evaluation protocols are not widely agreed upon in the field, and so there isn't a clear protocol to follow.

**Clarity**:

Generally, the paper is quite well-written. The architecture is well-motivated (particularly the loss in section 3.1) and the experimental setup (both evaluation metrics and choice of datasets) is well-argued.

**Questions**:
- did you perform a sensitivity analysis for the loss function? While the motivation for the loss function is extensively described, I am curious whether all terms are necessary.

**Minor comments** (did not affect the score in any way):
- use the following symbol twice for the left quotation marks in latex: `
- there are some minor typos/grammar/spacing errors.

**Summary Of The Paper:**

The authors introduce DISSECT: Disentangled Simultaneous Explanations via Concept Traversal. The method creates counterfactual explanations for which the concept increasingly influences the classifier's decision. Experimentally, the authors show that the proposed method generally performs better than existing methods on several aspects ("importance", "realism", "distinctness", "stability").

**Summary Of The Review:**

Overall, I would recommend an acceptance. The field of interpretability is becoming increasingly more important. The authors introduce a novel method, called DISSECT, which brings a meaningful contribution to the aforementioned field.  The method is novel (bringing a fresh perspective) and performs well empirically. I did not give an 8 as I still had some reservations about some aspects of the paper (as highlighted in the section before).

---

> ### Author Response · Authors · 2021-11-23
> **Author response**
>
> We thank the reviewer for their thoughtful review and for recognizing the novelty and impact of our work through multiple counterfactual generations, and our rigorous experiments.
>
> ## Evaluation:
>
> We have added **a new section to the appendix, A.3**, where we provide more details about the evaluation metrics. First, we provide evidence supporting these criteria as desirable qualities for explainability. Then, we summarize how these qualities have been measured in prior work. Finally, we explain when we can use prior formulation of these metrics as is, and when we need to modify them to make them applicable to the counterfactual explanation case. We justify how these formulations capture the desired criteria.
>
> Additionally, we have expanded realism metrics by adding Fréchet inception distance (FID). We have **added a new column to Tables 2-3 for the FID scores**.
>
> ## Sensitivity analysis:
> We have conducted a new experiment with different $\lambda_r$ values to showcase the importance of the disentangler in achieving successful disentanglement. We have clarified that **EPE-mod can be viewed as the ablated version of DISSECT** where $\lambda_r = 0$. All these results are summarized in **a new section in the appendix, A.6.**
>
> We have updated the paper to address typos and formatting errors.

---

> > ### Comment · Reviewer_1keG · 2021-11-29
> > **Thanks for your rebuttal**
> >
> > Thank you for your response and the changes to the paper. This clarifies some aspects.

---

### Official Review · Reviewer_jPGJ · 2021-11-02

**Correctness:** 3
**Technical Novelty And Significance:** 3
**Empirical Novelty And Significance:** 3
**Recommendation:** 6
**Confidence:** 4

**Main Review:**

Originality: The task is interesting, the proposed DISSECT is based on Explanation by Progressive Exaggeration (EPE) while proposing disentanglement modules. It is good to provide the pipeline comparison between EPE-mod and DISSECT, because they may look similar, which shows limited novelty.

Clarity: Overall is clear, however, some sentences are too long to follow: e.g., 1st paragraph in the Method section.

Significance and experiments:

1 Baselines method and evaluation setting are thorough, while it would be better to show ablation study and show the necessity of each component, especially the difference between EPE-mod and DISSECT.

2 I am curious about the generalization to a novel dataset, how to define the meaning of each concept that influences the classification?

3 For fidelity of the explanation, I wondered about the correctness of the concept. How to verify the correctness of the concepts? How to rank the importance of the concepts for an explanation?

4 For How to guarantee the provided CTs can be easily understood by humans? How to avoid confusing users when more concepts emerge?



**Summary Of The Paper:**

This paper proposed a generation-based explainability method, DISSECT, that generates Concept Traversals to provide a counterfactual explanation. The Concept traversals (CTs) are sequences of generated examples with increasing degrees of concepts' influence on the classifier's decision. The CTs are able to provide disentangled explanations for different concepts.

**Summary Of The Review:**

I am borderline for this paper in the current stage and I may increase my score if the author can solve the above concerns.

---

> ### Author Response · Authors · 2021-11-23
> **Author response**
>
> We thank the reviewer for their thoughtful comments and feedback.
>
> ## Pipeline comparison:
>
> We have **updated Figure 6** in the appendix A.2 to highlight the pipeline differences between EPE, EPE-mod, and DISSECT approaches. We have clarified that EPE-mod is the ablated version of DISSECT without the disentanglement regularizer. Since the novelty of our approach is the disentanglement piece, both from a practical application perspective and in the proposed approach, our aim is to leverage prior work as much as possible to limit the new added variables. As mentioned by reviewer WoTB, “to obtain multiple counterfactuals is novel and quite useful for multiple practical applications.” Additionally, as reviewer 1keG mentions, “the approach taken in this paper is very different from other counterfactual explanation methods. This is the first paper I've read that creates several explanations which increasingly rely on a concept. I think this aspect is very interesting and could be relevant in practice. For example, as highlighted in the paper, it can be used to depict how a mole turning from benign to malignant may progress.”
>
> ## Clarity:
> We have edited the text, especially the method’s section, to improve clarity.
>
> ## Ablation study:
> We have conducted a new experiment with different $\lambda_r$ values to showcase the importance of the disentangler in achieving successful disentanglement. We have clarified that **EPE-mod can be viewed as the ablated version of DISSECT** where $\lambda_r = 0$. All these results are summarized in a **new section in the appendix, A.6.**
>
> ## Generalization:
> We find two possible interpretations of the reviewer's question. The first is how can we assign names to concepts discovered by DISSECT in an unseen task? This interpretation overlaps with question 4 and touches on what concepts are understandable by humans. We will address this concern below, in “Understandability by humans” part of this response. We have reorganized the appendix to include a section on qualitative evaluation, A.5. In the **new section A.5.3**, we discuss the limitations of the current work and recommendations for future human-subject studies. Note that in this work, we have focused on case studies where we have control over the ground truth concepts of interests. DISSECT is still conducting unsupervised discovery of concepts. However, by procedurally generating the dataset or subsampling data, we are able to evaluate DISSECT more effectively.
>
> Our second interpretation of this question is, would DISSECT generalize to a new dataset? This is the case where we know the ground truth concepts in the new dataset, and would like to evaluate DISSECT’s ability to successfully discover them. We believe that our comprehensive experiments with 3 significantly different datasets already provide solid evidence for DISSECT’s generalization ability. For the future, we would like to apply DISSECT to more datasets and domains to further confirm this hypothesis.
>
> ## Fidelity of our explanations:
> The importance metrics included in the paper quantitatively measure the gradual increase of the target class’s posterior probability through a concept traversal. These metrics measure fidelity of explanations, or how truthful they are to the classifier-under-test. Table 1-3 provide aggregated importance scores over all the concepts. These scores can be provided per individual concept, as well. Consequently, they can be used to rank the importance of concepts. We have clarified this in Sec 4.2. Additionally, we have **added a new section to the appendix, A.7, where we provided the importance measures for individual concepts (Table 7).**
>
> Furthermore, our method by design supports fidelity by training the generator with respect to the classifier’s outcome. Note that $G(x, c(\alpha, k))$ is the image generated by perturbing $x$ with degree $\alpha$ of concept $k$. Equation 1 directly minimizes the log loss between the desired probability $\alpha$ and the predicted probability of the perturbed image $f(G(x, c(\alpha, k)))$. We have added a sentence to clarify this in the text.

---

> > ### Author Response · Authors · 2021-11-23
> > **Author response (cont'd)**
> >
> > ## Understandability by humans:
> > As mentioned in Section 6’s limitation, this technique does not guarantee finding all the concepts nor semantically meaningful concepts. We have reorganized the appendix to include a section on qualitative evaluation, A.5. In the **new section A.5.3**, we discuss the limitations of the current work and recommendations for future human-subject studies:
> >
> > While we provide several qualitative examples, further confirmation from human-subject studies to validate that CTs exhibit semantically meaningful attributes could strengthen our findings.
> >
> > The number of concepts to be discovered by DISSECT is a hyper-parameter that can be selected by the user. While distinctness and substitutability are designed to be globally evaluated across all concepts, importance, realism, and stability scores can be calculated for each concept. We hypothesize that ranking the discovered concepts based on these individual scores can benefit the users. In particular, it can help them focus on more salient concepts and not be overwhelmed by less informative concepts. Future user-studies are required to investigate these hypotheses.

---

> > ### Comment · Reviewer_jPGJ · 2021-11-29
> > **Thanks for your feedback and score increased**
> >
> > I have read the author's response and agree that overall they have addressed the concerns I had. I am changing my score to 6.

---

### Official Review · Reviewer_WoTB · 2021-11-02

**Correctness:** 4
**Technical Novelty And Significance:** 2
**Empirical Novelty And Significance:** 3
**Recommendation:** 6
**Confidence:** 3

**Main Review:**

The idea of using VAEs to generate multiple counterfactuals is novel. Authors have evaluated their work on 3 image datasets and a number of useful evaluation metrics and baselines. The results are promising.

Could you please comment on similarity/differences with this work:
https://arxiv.org/pdf/1905.12698.pdf



**Summary Of The Paper:**

In line with further work on counterfactual explanations, this work proposes to provide multiple counterfactual explanations in terms of VAE style disentangled latent variables. The idea of using VAEs to obtain multiple counterfactuals is novel and quite useful for multiple practical applications.

**Summary Of The Review:**

While there is work on diverse counterfactual explanations and different VAEs (DIP-VAE, Beta-VAE, etc), and work algorithmic recourse, this work seems to marry the two ideas of of providing multiple counterfactuals based on disentangles latent variables. This can be quite helpful in practise.

---

> ### Author Response · Authors · 2021-11-23
> **Author response**
>
> We thank the reviewer for their time, and for recognizing the value of our work in discovering multiple counterfactuals using an autoencoding approach. While the motivation of our work overlaps with Luss et al., we are taking a different approach in our proposed solution. We allow the classifier’s gradients  to flow through the generator during training. We focus on successfully training the generative model such that it does linear interpolations without additional optimization. However, Luss et al. use a sequential approach. They use a pre-trained generator. Then, find the trajectory on the learned embedding space using their proposed optimization method.
>
> One potential failure point for the latter approach is if the generative model only captures factors of variation that are independent of the classifier.  It could happen if factors relevant to the classifier relate to only a small part of the data manifold, and training a generative model without that knowledge, might not even capture that part of the manifold. Thus, finding a relevant trajectory in the generative model’s embedding space might not be successful.
>
> We have updated the related work section to include Luss et al. work.

---

### Decision · Program_Chairs · 2022-01-20

**Decision:**

Accept (Poster)

**Comment:**

This paper proposes a counterfactual explanation method, termed DISSECT, for image classification. While previous work is concerned with generating one single counterfactual, DISSECT aims to produce multiple counterfactuals, with each illustrating one possible way the class label could be altered. Intermediate images between the benign example and the counterfactuals are also generated to show how the decision boundary is crossed.
The reviewers find the idea novel, the presentation clear, and the empirical evaluation thorough.  However, there are concerns regarding whether the method will generalize to other domains because it relies on a strong generative model.  In addition, there is no human-subject study to show whether and how much the method really help an end-user.